# DISCOVERY AND EXPANSION OF
# NEW DOMAINS WITHIN DIFFUSION MODELS

## ABSTRACT

In this work, we study the generalization properties of diffusion models in a few-shot setup, introduce a novel tuning-free paradigm to synthesize the target out-of-domain (OOD) data, showcase multiple applications of those generalization properties, and demonstrate the advantages compared to existing tuning-based methods in data-sparse scientific scenarios with large domain gaps. Our work resides on the observation and premise that the theoretical formulation of denoising diffusion implicit models (DDIMs), a non-Markovian inference technique, exhibits latent Gaussian priors independent from the parameters of trained denoising diffusion probabilistic models (DDPMs). This brings two practical benefits: the latent Gaussian priors generalize to OOD data domains that have never been used in the training stage; existing DDIMs offer the flexibility to traverse the denoising chain bidirectionally for a pre-trained DDPM. We then demonstrate through theoretical and empirical studies that such established OOD Gaussian priors are practically *separable from* the originally trained ones after inversion. The above analytical findings allow us to introduce our novel tuning-free paradigm to synthesize new images of the target unseen domain by *discovering* qualified OOD latent encodings within the inverted noisy latent spaces, which is *fundamentally different* from most existing paradigms that seek to modify the denoising trajectory to achieve the same goal by tuning the model parameters. Extensive cross-model and domain experiments show that our proposed method can expand the latent space and synthesize images in new domains via frozen DDPMs without impairing the generation quality of their original domains.

## 1 INTRODUCTION

Generalization ability, which enables the model to synthesize data from various domains, has long been a challenge in deep generative models. The current research trend focuses on leveraging larger models with more training data to facilitate improved generalization. The popularity of recent large-scale models such as DALLE-2 (Ramesh et al., 2022), Imagen (Ho et al., 2022a) and StableDiffusion (Rombach et al., 2022) have demonstrated the impressive and promising representation abilities of the state-of-the-art (SOTA) diffusion generative models when trained on an enormous amount of images such as LAION-5B (Schuhmann et al., 2022). However, brute-force scaling up is not a panacea and does not fundamentally solve the generalization challenge. In other words, for data domains that remain sparse in those already giant natural image datasets, such as the astrophysical observation and simulation data, even the SOTA models fail to synthesize data suitable for rigorous scientific research purposes, as those data usually follow physical distributions that are dramatically distinguishable from natural images in computer vision, illustrated in the Fig. 1. In addition, scaling up requires exhaustive resources, severely limiting the number of research groups that are able to participate and contribute to the work, and consequently hindering research progress. Given the concerns above, our work focuses on *studying the generalization ability in a few-shot setup*, where a pre-trained diffusion generative model and a small set of raw data different from its training domain are provided, with the ultimate objective of generating new data samples for the target OOD domain.

The fundamental challenge of domain generalization in deep generative models lies in learning a mapping function that accurately captures the structure of a high-dimensional, irregular data space with an unknown distribution. This is even statistically difficult with large datasets when using non-parametric machine learning methods, a phenomenon known as the "curse of dimensionality," and

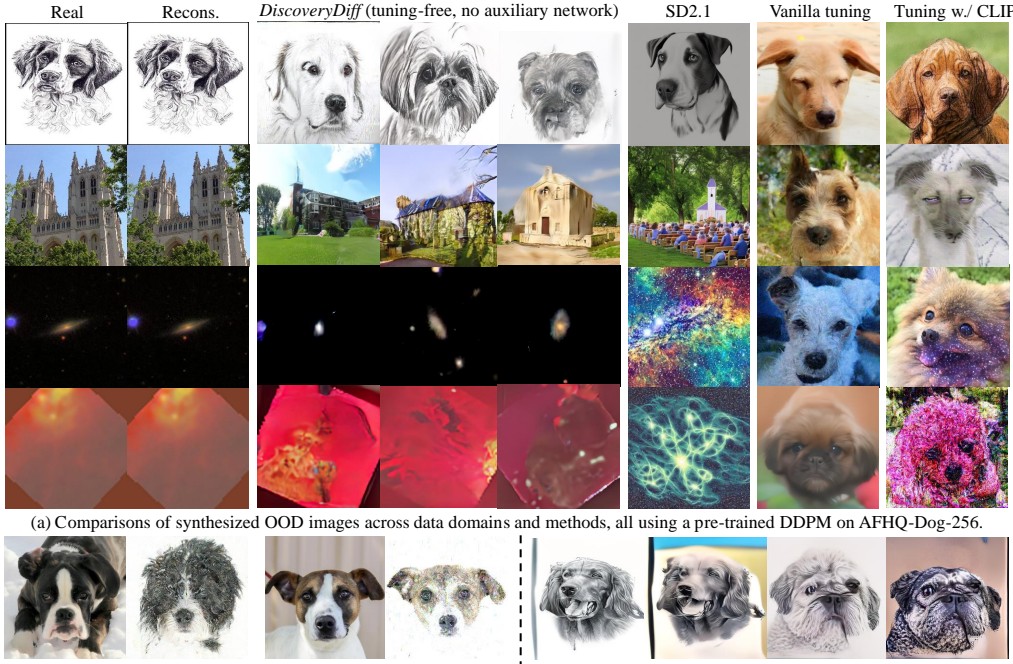

Real    Recons.    *DiscoveryDiff* (tuning-free, no auxiliary network)    SD2.1    Vanilla tuning    Tuning w./ CLIP

(a) Comparisons of synthesized OOD images across data domains and methods, all using a pre-trained DDPM on AFHQ-Dog-256.

(b) In additional to unconditional synthesis, our proposed method can also achieve bi-directional domain transfers (e.g.. sketch - RGB).

Figure 1: **Our proposed tuning-free *DiscoveryDiff* method for synthesizing OOD data in a few-shot setup.** Using a pre-trained DDPM on AFHQ-Dog (Choi et al., 2020) RGB images as an example, we can well reconstruct arbitrary unseen images across domains covering sketch images, RGB images in other classes (e.g., outdoor churches), and even astrophysical data. By leveraging such representation abilities from the frozen model, we can establish OOD latent priors through deterministic inversion (Song et al., 2021) and our effective latent encoding discovery mechanism to achieve applications such as *(a)* unconditional synthesis and *(b)* bi-directional domain transfer, without modifying original or learning additional parameters. In contrast, tuning the same model usually fail to generate dramatically different data domains.

becomes even more challenging in few-shot scenarios. As shown in Fig. 1, directly fine-tuning the entire diffusion model, either with raw images or with additional semantic guidance such as CLIP loss (Radford et al., 2021) fails to transfer the synthesis domain from the pre-trained model's original data (e.g., dog images) to new data domains, especially when the domain gap between the trained one and the target one is large (e.g., dog-to-church images or astrophysical data). In addition to the unsatisfactory performance, tuning-based methods have several other drawbacks, for instance, modifying parameters adversely affects synthesis quality in the model's originally trained domain. The tuning cost is also entirely dependent on the pre-trained model, which can be quite high given the well-known expensive training for diffusion models. Very few existing works have explicitly investigated this task. Most recent works focus on downstream applications to control pre-trained models in the context of data editing, manipulations, and styling (Kim et al., 2022; Kwon et al., 2023; Zhao et al., 2022; Zhang et al., 2023), which brings a certain level of generalization ability of the original model to different but still related domains (e.g., style transfer). In this work, unlike the existing trajectory-tuning paradigm, we introduce a heuristic approach featuring a novel **tuning-free** paradigm that achieves domain generalization by sampling latent encodings of the unseen target domain within the latent spaces of pre-trained diffusion models. Our core idea is to *discover the corresponding OOD latent encodings* and denoise them through deterministic trajectories in DDIMs (Song et al., 2021), as depicted in Fig. 2. Intuitively, our approach *leverages the intrinsic mathematical properties of the generative dynamics of diffusion models to reduce the heavy data dependence* typically required by conventional non-parametric distribution modeling methods.

Our work resides on the observation and premise that the theoretical formulation of denoising diffusion implicit models (DDIMs) (Song et al., 2021), a non-Markovian and deterministic inference technique, exhibits latent Gaussian priors independent from the parameters of trained denoising dif-

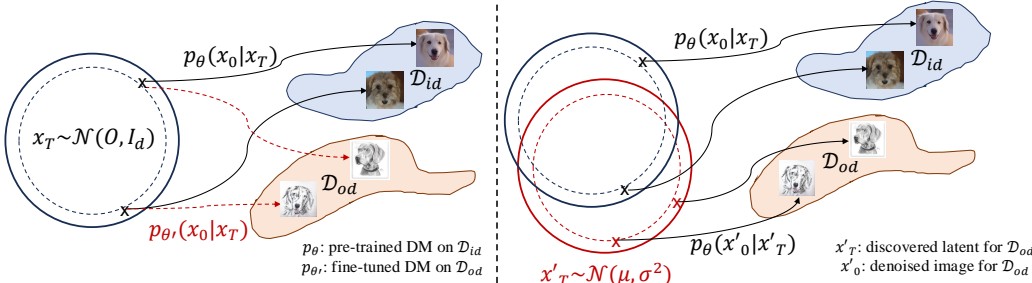

Figure 2: **Illustration of the trajectory-tuning based paradigm (left) and our proposed latent-discovery based paradigm (right) for OOD image synthesis with diffusion models.** Given a pre-trained DM $p_\theta$ on images from domain $\mathcal{D}_{id}$, most existing methods seek to finetune the generation trajectories $p_{\theta'}$ to synthesis data in a new domain $\mathcal{D}_{od}$. In contrast, we propose to discover unseen latent encodings to achieve the same goal via the frozen model $p_\theta$ by expanding the latent spaces.

fusion probabilistic models (DDPMs) (Ho et al., 2020). In addition, DDIMs also provide a tractable way of traversing bidirectionally along the generation trajectory, which further demonstrates that DDPMs trained on single-domain images already have sufficient representation ability to reconstruct images from arbitrary unseen domains from the inverted OOD latent encodings, as shown in Fig. 1. The results from the arbitrary image reconstruction test via the deterministic inversion[1] (i.e., diffusion direction) and denoising (i.e., reserve direction) suggest that *an inverse direction* to solve our task objective: by *identifying additional qualified OOD latent encodings* based on the established priors from a limited set of OOD images, we can synthesize unseen images without the need to adjust the original model parameters. While the reconstruction ability is not entirely novel and has been widely adopted for a line of downstream works that seek to control the generative output for tasks such as image editing (Kwon et al., 2023; Zhu et al., 2023a) and customization (Yang et al., 2024), leveraging such latent representation capacity for new OOD domain generalization further requires two critical prerequisites that have remained underexplored: a (relatively) known OOD prior, and a clear separability between the target OOD domain and originally trained domain both in the latent spaces and denoising trajectories. From a high-level perspective, the first prerequisite enables us to achieve the general unconditional synthesis, thus reducing the reliance on a given known data point in contrast to specific downstream tasks such as image editing. As for the second prerequisite, it is vital to generate high-quality *OOD* images to avoid the "mode interference" issue in Sec. 2.4, which is also qualitatively illustrated in Fig. 3.

Based on our in-depth analytical studies in Sec. 2, we introduce our paradigmatic design that achieves unseen domain synthesis in a tuning-free manner by discovering additional latent OOD encodings based on the inverted priors, as described in Sec. 3. Conceptually inspired by recent works that seek to manipulate the in-domain semantic attribute direction for data editing (Zhu et al., 2023a; Baumann et al., 2024), the key idea of our proposed OOD sampling method focuses on identifying potential latent directions by leveraging the geometric properties of the inverted OOD domains as additional domain-specific priors. The key takeaway from this part echoes our earlier analysis, demonstrating that in this latent discovery paradigm, the core technical challenge arises from the tendency of discovered OOD samples to be interfered with and captured by the original trained domain. This further underscores its distinguishable nature with intuitive tuning-based methods, where smaller data domain gaps are preferable for achieving better generalization performance.

Finally, we conduct extensive downstream experiments to demonstrate the effectiveness of our proposed paradigm. As all of our analysis are conducted within the formulation-level, the findings generalize across different DDPM variants, such as vanilla DDPMs (Ho et al., 2020) and improved DDPMs (Nichol & Dhariwal, 2021). Notably, our experiments are carefully designed to represent **an increasing level of domain gaps** and to showcase versatile application scenarios. For the target OOD domains, we test with RGB images in different classes, sketch images (Wang et al., 2019a), scientific images (Willett et al., 2013), and even non-image astrophysical radiation emission data (Xu et al., 2023a). In addition to the quantitative and qualitative evaluation of natural images

---

[1]Inversion refers to the process of converting raw data to noisy encodings in the literature of generative modeling (Xia et al., 2022), which can also be understood as a diffusion process in the context of DMs.

widely adopted in the computer vision community, we also involve astrophysicists to independently assess the quality of generated data in comparison to the astrophysical simulations. These comprehensive evaluations further reinforce our findings.

Overall, we hope to provide insights into a novel perspective for understanding the domain generalization abilities of diffusion models through a challenging few-shot scenario in this work, and to shed light on potential directions for broader interdisciplinary applications. Our main contributions are summarized below:

- We present an in-depth and comprehensive analytical study to investigate the OOD latent distributions and reveal their separability concerning the models' originally trained domains from both theoretical and empirical perspectives.

- We introduce our *DiscoveryDiff* method, featuring a tuning-free paradigm that aims to discover additional OOD latent encodings to expand the synthesis domains of frozen DDPMs.

- We conduct extensive experiments with a wide range of diffusion models and datasets, showcasing the applicable tasks and demonstrating the performance superiority compared to tuning-based methods in dramatically different domains with astrophysical data.

## 2 DOMAIN DISCOVERY AND EXPANSION OF DDPMs

This section presents our problem formulation under the few-shot scenario and our analytical studies on the generalization properties of pre-trained DDPMs in the context of deterministic trajectories from theoretical and empirical perspectives. The *high-level takeaway* from our in-depth study is that the inverted OOD samples establish Gaussian separable from the trained ID prior. The key technical challenge is to find qualified OOD latent free from the *"mode interference"*, which is distinguishable from the common understanding in tuning-based designs.

### 2.1 PROBLEM FORMULATION AND NOTATIONS

The general goal of DDPMs is similar to most previous generative models, which is to approximate an implicit data distribution $q(\mathbf{x}_0)$ with a learned model distribution $p_\theta(\mathbf{x}_0)$, as well as providing an easy-to-sample proxy (e.g., standard Gaussian). For the conventional unconditional generation process to synthesize $\mathbf{x}' \sim q(\mathbf{x}_0)$, we first draw $\mathbf{x}'_T \sim \mathcal{N}(0, I_d)$ and obtain $\mathbf{x}' = p_\theta(\mathbf{x}'_{0:T})$.

In this work, given a DDPM $p_\theta$ trained on images $\mathbf{x}_0$ from a domain $\mathcal{D}_{id}$, we aim to study the behavior of $q(\mathbf{x}_{od,1:T}|\mathbf{x}_{od})$ on other unseen domain $\mathcal{D}_{od}$ using $N$ data samples $\mathbf{x}_{od} \in \mathcal{D}_{od}$, with $N$ to be a relatively smaller number compared to the usual requirements to train a diffusion model from scratch. The ultimate goal is to obtain new data samples $\mathbf{x}'_{od} \in \mathcal{D}_{od}$, with the assumption to discover $\mathbf{x}'_{T,od} \sim q(\mathbf{x}_{od,T}|\mathbf{x}_{od})$, such that $\mathbf{x}'_{od} = p_\theta(\mathbf{x}'_{od,0:T})$.

As for notations, we use $p_s$ and $p_i$ to represent the stochastic (Ho et al., 2020) and deterministic (Song et al., 2021) generation processes, respectively. We omit $\theta$ as we use frozen pre-trained models. In addition, we use the hyper-parameter $\eta$ (Song et al., 2021) to characterize the degree of stochasticity in the generative process, with $\eta = 1$ for $p_s$ and $\eta = 0$ for $p_i$. At intermediate stochastic levels, we adopt the notation $p_{\eta=k}$ with $k$ equals a constant between 0 and 1. Similar to existing literature, $T$ denotes the total diffusion steps. We use $\mathcal{X}_t$ to represent the latent (noisy) spaces formed by $\mathbf{x}_t$ along denoising.

### 2.2 REPRESENTATION ABILITY OF LATENT SPACES IN DETERMINISTIC DIFFUSION

A diffusion generative model, trained even on a single-domain small dataset (*e.g.*, dog faces), already has sufficient representation ability to accurately reconstruct arbitrary unseen images (*e.g.*, sketch, church, and astrophysical data), as shown in the second column of Fig. 1. The reconstruction ability is subject to the deterministic inversion and denoising trajectories (Song et al., 2021), which can be considered as a special case of vanilla stochastic process (Ho et al., 2020). The findings above suggest that: with a good mapping approximator (i.e., pre-trained DDPM) and proper tool (i.e., deterministic trajectories with DDIMs), its intermediate latent spaces already have sufficient representation ability for arbitrary images, which opens up the possibility to leverage DDPMs

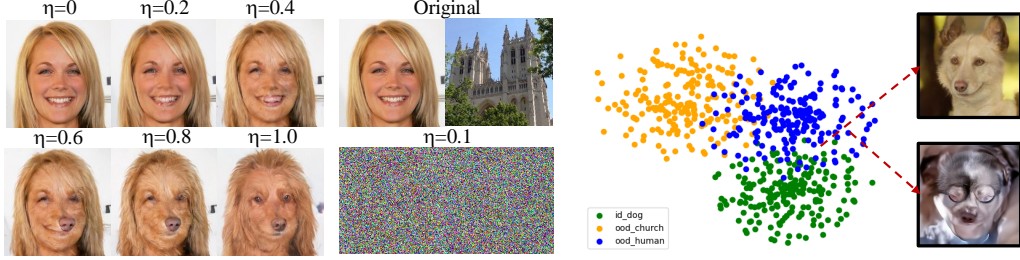

(a) Mode interference as η increases.  (b) Test w./ untrained DMs.  (c) T-SNE of inverted latent encodings and failure cases.

Figure 3: **Various visualizations of "mode interference" and latent generalization properties.** Given an example of base DDPMs trained on dogs. *(a):* An interfered image of human faces gradually becomes similar to its original trained domain as the denoising trajectory shifts from deterministic ($\eta = 0$) to stochastic ($\eta = 1$). *(b):* Untrained DMs can also reconstruct arbitrary images via DDIMs, but such case only establishes bijective mapping w/o actual generalization abilities. *(c):* Failure cases happen when sampled latent OOD encodings are captured by the model's original probabilistic concentration mass, leading to generated target OOD data become "in-distribution".

for synthesizing images from new domains *without tuning* the model parameters. The quantitative evaluation of the reconstruction results is in Tab. 3 in the Appendices.

The core idea in the context of deterministic non-Markovian DDIMs (Song et al., 2021) is to consider a family of $\mathcal{Q}$ of inference distributions, indexed by a real vector $\sigma \in \mathbb{R}_{\geq 0}^T$:

$$q_\sigma(\mathbf{x}_{1:T}|\mathbf{x}_0) := q_\sigma(\mathbf{x}_T|\mathbf{x}_0) \prod_{t=2}^{T} q_\sigma(\mathbf{x}_{t-1}|\mathbf{x}_t, \mathbf{x}_0), \tag{1}$$

where $q_\sigma(\mathbf{x}_T|\mathbf{x}_0) = \mathcal{N}(\sqrt{\alpha_T}\mathbf{x}_0, (1-\alpha_T)\mathbf{I})$ and $\forall t > 1$,

$$q_\sigma(\mathbf{x}_{t-1}|\mathbf{x}_t, \mathbf{x}_0) = \mathcal{N}(\sqrt{\alpha_{t-1}}\mathbf{x}_0 + \sqrt{1 - \alpha_{t-1} - \sigma_t^2} \cdot \frac{\mathbf{x}_t - \sqrt{\alpha_t}\mathbf{x}_0}{\sqrt{1-\alpha_t}}, \sigma_t^2\mathbf{I}), \tag{2}$$

with $\alpha_{1:T} \in (0, 1]^T$ is a decreasing sequence that parameterizes Gaussian transition kernels.

### 2.3 MODEL PARAMETER INDEPENDENT PROPERTIES: GAUSSIAN PRIORS

While this deterministic line of works was initially proposed to accelerate the vanilla ancestral sampling, later studies (Kwon et al., 2023; Zhu et al., 2023a; Yang et al., 2024) revealed that the deterministic diffusion can be used as a tractable and lossless way for conducting data inversion to achieve downstream data editing and customization. However, in addition to this deterministic properties as the tool for inversion and denoising the unseen images, we note the following property offered by its original formulation but has been yet under-exploited. The takeaway message is: *In theory*, the inverted latent encodings also establish Gaussian priors as presented in Lemma 2.1.[2]

**Lemma 2.1.** *For $q_\sigma(\mathbf{x}_{1:T}|\mathbf{x}_0)$ defined in Eqn. 1 and $q_\sigma(\mathbf{x}_{t-1}|\mathbf{x}_t, \mathbf{x}_0)$ defined in Eqn. 2, we have:*

$$q_\sigma(\mathbf{x}_t|\mathbf{x}_0) = \mathcal{N}(\sqrt{\alpha_t}\mathbf{x}_0, (1-\alpha_t)\mathbf{I}). \tag{3}$$

As also mentioned in Song et al. (2021), one can derive Lemma 2.1 by assuming for any $t \leq T$, $q_\sigma(\mathbf{x}_t|\mathbf{x}_0) = \mathcal{N}(\sqrt{\alpha_t}\mathbf{x}_0, (1-\alpha_t)\mathbf{I})$ holds, if:

$$q_\sigma(\mathbf{x}_{t-1}|\mathbf{x}_0) = \mathcal{N}(\sqrt{\alpha_{t-1}}\mathbf{x}_0, (1-\alpha_{t-1})\mathbf{I}), \tag{4}$$

and then prove the statement with an induction argument for $t$ from $T$ to 1, since the base case ($t = T$) already holds by definition. Proof details can be found in Appendix C. We note that derivations are completed in the forward diffusion direction (i.e., the direction from data to latent spaces), and make no modification to the trained model. This sets the *primary theoretical grounding* for estimating the latent distributions as Gaussian in a model parameter-independent manner.

---

[2]However, in practice, due to the fact that pre-trained DMs themselves are function approximators, the samples after inversion do not establish *perfect* Gaussians but rather approximations, echoing Sec. 3.

2.4 DATA DEPENDENT PROPERTIES: MODE INTERFERENCE AND SEPARABILITY

In the literature of GANs-based generative models (Goodfellow et al., 2014), "mode collapse" is a common issue that describes the training failure when generated images tend to be very similar given randomly sampled starting encodings from the Gaussian prior. Within the context of diffusion models in our work, we explicitly reveal a phenomenon analog to the "mode collapse" in GANs, which we refer to as *"mode interference"*, as qualitatively illustrated in Fig. 3 (a).

Intuitively, *"mode interference"* describes the case when the denoised images fall into the model's original training domain $\mathcal{D}_{id}$ instead of the target unseen domain $\mathcal{D}_{od}$ due to the prior interference in the latent spaces. Specifically, when we sample directly from the standard Gaussian to obtain a latent encoding $\mathbf{x}_T \sim \mathcal{N}(\mathbf{0}, \mathbf{I}_d)$, then the denoised image will surely fall into the original training domain $\mathbf{x}_0 \in \mathcal{D}_{id}$ with $\mathbf{x}_0 \sim p_\theta(\mathbf{x}_0)$, which is the vanilla generation process of a trained DDPM. However, it contradicts our task objective to synthesize images $\mathbf{x}_0' \in \mathcal{D}_{od}$. As illustrated in Fig. 2, since we are denoising the latent encoding via deterministic trajectories $p_i$, the remaining critical technical challenge to generate $\mathbf{x}_0'$ is to find additional qualified latent encoding $\mathbf{x}_T'$ *free from the interference* of the ID Gaussian mode in the sampling stage.

Notably, a key precondition to achieving the effective OOD latent sampling is that the established OOD prior mode *should be separable* from the ID Gaussian prior mode (i.e., a standard Gaussian). Otherwise, the denoised image would fall into the training domain as in Fig. 3 (c). The separability is further supported and validated by our empirical verification below in Sec. 2.5.

2.5 ANALYTICAL EXPERIMENTS

We show empirical verification from multiple perspectives to support our model parameter independent and data dependent properties described in Sec. 2.3 and Sec. 2.4.

**Geometrical Properties of Gaussians.** We leverage the geometrical measurements established of the high-dimensional studies in mathematics (Blum et al., 2020), as additional empirical support for the Gaussian priors in Sec. 2.3. Specifically, we compute several geometric metrics, including the pair-wise angles (angles formed by three arbitrary samples), sample-to-origin angles (angles formed by two arbitrary samples and the origin), pair-wise distance (euclidean distance between two arbitrary samples) and distance between OOD and ID Gaussian centers, with results in Tab. 1.

The characteristics mentioned above are typical geometric properties of isotropic high-dimensional Gaussians (Blum et al., 2020). Notably, three randomly sampled points from a high-dimensional Gaussian distribution almost surely form an approximately equilateral triangle, with pairwise angles close to $60°$, and are nearly orthogonal to each other, as reflected by the $90°$ sample-to-origin angles shown in the first and second rows of Tab. 1, respectively. However, it is important to note that while these geometric properties are common in high-dimensional Gaussians, they are sufficient but not necessary conditions for identifying such distributions. In other words, these geometric properties alone are not enough to infer the underlying distribution without additional prior information. More details about the geometric properties are in Appendix C.

**Mode Separability.** As revealed by our analysis in Sec. 2.4, the separability between ID and OOD Gaussian modes is critical for synthesizing target unseen domain images without modifying the model parameters and for avoiding the "mode interference". We further provide validation from the statistical and learning-based classifier perspectives to support the separability claim.

*Statistical Validation.* The separability of high-dimensional Gaussians follows Lemma 2.2 (Blum et al., 2020), which states that spherical Gaussians can be relaxingly separated by $\Omega(d^{\frac{1}{4}})$, or even $\Omega(1)$ with more sophisticated algorithms. In other words, for a DDPM trained on $256 \times 256$ images with dimensionality $d = 3 \times 256 \times 256$, ID and OOD modes can be well separated and avoid interference given a distance larger than $d^{\frac{1}{4}} \approx 21$, which is further validated by the empirical distance between centers, listed in the forth row of Tab. 1. More details about Lemma 2.2 are in Appendix C.

**Lemma 2.2.** *Mixtures of spherical Gaussians in $d$ dimensions can be separated provided their centers are separated by more than $d^{\frac{1}{4}}$ distance (i.e., a separation of $\Omega(d^{\frac{1}{4}})$).*

*Classifier Validation.* Another empirical perspective to validate the separability between modes in the latent spaces is using the classifiers as in existing literature (Shen et al., 2020; Zhu et al., 2023a).

Table 1: **Geometric properties of inverted ID and OOD latent encodings at step T.** The results are computed based on 1K sample pairs. We report the mean and std for each geometric measurement to ensure the statistical significance. The base model is trained on AFHQ-Dog-256 (Choi et al., 2020).

| $\mathcal{D}$ | Dog (ID) | Sketch (O) | Human (O) | Church (O) | Astro. Galaxy (O) | Astro. Radiation (O) |
|---|---|---|---|---|---|---|
| Pair-Angle | 60.0±0 | 60.0±0 | 60.0±0.1 | 60.0±0.1 | 60.0±0.1 | 60.0 ± 0 |
| Angle-Origin | 89.7±0.01 | 89.1±0.01 | 89.8±0.01 | 89.7±0.01 | 89.1±0.01 | 87.6 ±0.03 |
| Pair-Distance | 607.4±0.01 | 622.5 ± 0.02 | 619.3±0.07 | 620.4±0.02 | 612.37±0.05 | 609.13 ± 0.1 |
| Center-Distance | - | 58.0 | 31.8 | 30.7 | 54.7 | 80.6 |
| Clf. Acc. | - | 0.96 | 0.99 | 0.99 | 1.0 | 1.0 |

Specifically, a linear classifier such as SVMs (Hearst et al., 1998) can be fitted to test the separability between ID and OOD encodings in the latent spaces. In our analytical experiments, we fit SVMs on 1K inverted ID and OOD samples following the 7:3 training-testing ratio, and report the test accuracy in Tab. 1. As additional clarification, the classification results are obtained with the test on the latent space $\mathcal{X}_T$. Our rationale behind the choice of $T$ corresponds to the recent findings of DMs (Zhu et al., 2023a; Yang et al., 2024), which indicates that $\mathcal{X}_T$, as the departure latent space, has the largest probabilistic support for the trained domain. In other words, if the latent ID and OOD modes can be separated in $\mathcal{X}_T$, they can be separated more easily in other $\mathcal{X}_t$, for $t = \{T - 1, ..., t, ...1\}$.

As shown in Tab. 1, while performing the binary classification task on the inverted latent encodings, a simple linear classifier can well separate ID and OOD domains, which further validates the latent modes are separable. In addition, we also observe that while the unseen images (e.g., human faces) are visually more similar to the trained domains (e.g., dogs), the inverted latent encodings *inherit such similarity*, making those unseen domains *more difficult to be separated* from the trained mode, and subsequently cause extra generation difficulties for those target domains via our tuning-free paradigm (see Sec. 3). We note this is distinguishable from previous tuning-based generalization works (Zhou et al., 2020; 2021; Wang et al., 2019b), which believe that it is usually easier to generalize model abilities to unseen domains similar to the trained ones, further validated in Sec. 4.

Our findings on the OOD mode separability also align with another thread of recent works that investigate pre-trained DDMs for discriminative tasks like classification and segmentation (Li et al., 2023; Clark & Jaini, 2023; Prabhudesai et al., 2023), where they reveal that diffusion models *generalize better to classifying out-of-distribution images*.

## 3 DISCOVERY-BASED OOD SYNTHESIS

Following our extensive analysis, we note that the methodological challenges in this work can be disentangled into two key points: sample qualified latent encodings from the OOD prior, and avoid the mode interference. Sampling from the inverted high-dimensional OOD priors, however, is an open and non-trivial challenging task given the theory-practice gap,[3] and points to multiple possible directions of solutions. In fact, high-dimensional Gaussian estimation itself remains a challenging research topic, especially in a multi-variant case (Zhou et al., 2011; Bai & Shi, 2011). While we present our proposed latent sampling method below, we have experimented with *many other methods* that may not yield the best performance, such as vanilla Gaussian sampling and MCMC, with detailed discussions in Appendix D.

**Few-Shot Latent References.** Having obtained $\mathbf{x}od, T$ from the $N$ raw images $\mathbf{x}_{od}$, it might seem intuitive to sample directly from the estimated Gaussian distribution $\mathcal{N}(\mu_{est}, \sigma^2_{est})$ based on these inverted OOD latent encodings. However, this approach is *empirically insufficient* to avoid mode interference because, in practice, the inversion does not yield a perfect Gaussian distribution, even for well-trained in-domain cases (Zhu et al., 2023a). To establish a more precise starting point in the OOD latent space, we propose using the inverted samples $\mathbf{x}_{od,T}$ as the initial point for navigating the subsequent sampling process. Furthermore, this concept of latent references can be flexibly adapted to other downstream applications, where the initial latent encoding is determined by the task setup, such as in the style translation between GRB and sketch images, as illustrated in Fig. 1(b).

**Latent Direction for Target Samples.** While the inverted latent sample $\mathbf{x}_{od,t}$ serves as a known starting point, we continue to mine more unknown latent samples that shall lead to new denoised

---

[3]As all the generative models can be considered as function approximators between the sampling prior and the implicit data distributions with certain error levels.

---

**Algorithm 1** Domain Discovery and Expansion within DDPMs

---

**Input:** $N$ raw images $\mathbf{x}_{od}$ from the target unseen domain $\mathcal{D}_{od}$, a pre-trained DDPM $p$ for domain $\mathcal{D}_{id}$, target latent step $t$ for sampling ($t$ is a pre-defined hyper-parameter discussed in Sec. 4.2).

**Output:** images of the unseen target $\mathbf{x}'_{od} \in \mathcal{D}_{od}$

*// Step 1: get the inverted OOD encodings $\mathbf{x}_{od,t}$*

Define $\{\tau_s\}_{s=1}^{S_{inv}}$ s.t. $\tau_1 = 0, \tau_{S_{inv}} = t$

**for** $i = 1, 2, ..., N$ **do**

   **for** $s = 1, 2, ..., S_{inv} - 1$ **do**

      $\epsilon \leftarrow p(\mathbf{x}^i_{od,\tau_s}, \tau_s)$

      $\mathbf{x}^i_{od,\tau_{s+1}} = \sqrt{\alpha_{\tau_s}}\mathbf{x}^i_{od,\tau_s} + \sqrt{1 - \alpha_{\tau_s}}\epsilon$

   **end for**

   Save the OOD latent $\mathbf{x}^i_{od,\tau_{S_{inv}}}$ as $\{\mathbf{x}_{od,t}\}^{i=1,...,N}$

**end for**

*// Step 2: find new OOD encodings $\mathbf{x}'_{od,t}$*

$\mu_{est} \leftarrow Mean(\{\mathbf{x}^1_{od,t}, \mathbf{x}^2_{od,t}, ..., \mathbf{x}^N_{od,t}\}), \sigma^2_{est} \leftarrow Var(\{\mathbf{x}^1_{od,t}, \mathbf{x}^2_{od,t}, ..., \mathbf{x}^N_{od,t}\})$

$\bar{\mathbf{x}}_{od,t} \sim \mathcal{N}(\mu_{est}, \sigma^2_{est}), \mathbf{x}_{od,t} \sim \{\mathbf{x}_{od,t}\}^{i=1,...,N}$

$\mathbf{x}'_{od,t} \leftarrow \mathsf{Slerp}(\mathbf{x}_{od,t}, \bar{\mathbf{x}}_{od,t})$

**If** Pass the geometric optimization in Algo. 2

$\mathbf{x}'_{od} \leftarrow p(\mathbf{x}'_{od,t}, t)$ *// Step 3: get denoised $\mathbf{x}'_{od} \in \mathcal{D}_{od}$*

---

images $\mathbf{x}'_{od}$. Specifically, inspired by several recent works in image editing through latent direction guidance (Zhu et al., 2023a; Baumann et al., 2024), we deploy the samples $\bar{\mathbf{x}}_{od,t} \sim \mathcal{N}(\mu_{est}, \sigma^2_{est})$ drawn from the estimate Gaussian as the ultimate latent directions, and obtain samples $x'_{od,t}$ along the spherical interpolation (slerp) (Shoemake, 1985) between $\mathbf{x}_{od,t}$ and $\bar{\mathbf{x}}_{od,t}$. The rationale behind the spherical interpolation comes from the fact that the probabilistic concentration mass of a high-dimensional Gaussian is mainly centered around a thin annulus around the equator (Blum et al., 2020). In the meantime, it is critical for discovered latent samples to stay within the area of high probabilistic concentration mass to ensure denoised data with high quality.

**Geometrical Optimization.** So far we have localized a trajectory with intermediate samples $\mathbf{x}'_{od,t} \in \mathsf{Slerp}(\mathbf{x}_{od,t}, \bar{\mathbf{x}}_{od,t})$ connecting two samples $\mathbf{x}_{od,t}$ and $\bar{\mathbf{x}}_{od,t}$ in this latent OOD space. To further improve the quality of our discovered latent samples, we leverage the geometric properties as domain-specific information to optimize the latent samples we have obtained from the previous step as additional criteria to avoid mode interference. Specifically, we can reject a fraction of initial samples via the angles and distances as shown in Tab. 1 by setting pre-defined tolerance ranges $\omega$.

**Overall Algorithm.** The overall pipeline of our proposed method includes the following major steps: raw image inversion via $p_i$, geometric property computation, latent sampling and optimization, and deterministic denoising via $p_i$, as shown in Algo. 1. More details in Appendix D.

## 4 DOWNSTREAM APPLICATION EXPERIMENTS

### 4.1 EXPERIMENTAL SETUP

**Model Zoos and Datasets.** We adopt multiple pre-trained DDPMs on different single domain datasets as our base models for experiments [4]: improved DDPM (Nichol & Dhariwal, 2021) trained on AFHQ-Dog (Choi et al., 2020), and DDPM (Ho et al., 2020) trained on CelebA-HQ (Karras et al., 2017), LSUN-Church (Yu et al., 2015), and LSUN-Bedroom (Yu et al., 2015). Each model generates images in the original resolution of $256 \times 256$, resulting in a total dimensionality of the latent spaces $d = 256 \times 256 \times 3 = 196,608$.

In addition to the above commonly used natural image datasets, we further experiment with the ImageNet-Sketch (Wang et al., 2019a) and two astrophysical datasets to cover a wide range of domain differences and to showcase the application scenarios. For sketch images, we select the subset

---

[4]Our proposed generalization analysis also holds for LDMs, however, the downstream applicable scenarios and performance induce a nuanced variance, as briefly discussed in our Appendix.

Table 2: **General quality evaluation in cross model and domain setup.** We report the FID scores (↓) for natural image domains and the Mean Opinion Scores (MOS) (↑) from subjective evaluations for astrophysical data. *Ref.* denotes if a raw image is provided as the starting point. High FID scores indicate tuning based methods *hardly work* given similar resource budget, more examples in Fig. 5.

| Methods | Ref. | Dog | CelebA | Church | Bedroom | Sketch | Galaxy | Radiation |
|---|---|---|---|---|---|---|---|---|
| Vanilla tuning | ✗ | 213.6±4.8 | 229.7±4.3 | 192.5±3.7 | 191.1±4.0 | 298.4±5.2 | - | - |
| CLIP tuning | ✗ | 204.1±4.2 | 218.7±4.0 | 196.4±3.6 | 193.2±4.1 | 257.7±4.8 | - | - |
| CLIP tuning | ✓ | 140.4±3.7 | 126.9±3.9 | 142.2±4.0 | 145.1±3.8 | 136.8±3.5 | 1.84±0.97 | 1.35±0.74 |
| Ours w/o Geo. Opt. | ✓ | 133.4±3.2 | 129.1±2.9 | 115.3±3.8 | 116.4±3.5 | 124.8±3.9 | - | - |
| Ours (tuning-free) | ✓ | **117.7**±3.6 | **114.4**±3.8 | **103.8**±3.0 | **105.6**±3.4 | **98.7**±3.6 | **2.88**±0.93 | **1.52**±0.80 |

of data with dog class labels. Specifically, we adopt the GalaxyZoo (Willett et al., 2013) and the radiation simulation data (Xu et al., 2023a), the latter has been investigated using DMs for prediction purposes. Details about those astrophysical datasets, their scientific interpretations, and evaluations are included in the Appendix E for interested readers, which differs from the usual interpretation and evaluation of natural images.

**Comparable Methods.** We mainly compare the performance with different tuning based methods on diffusion models. (1) *Basic baselines*: vanilla fine-tuning with the classic variational lower bound loss and reconstruction loss as proposed in DDPMs (Ho et al., 2020); (2) *Strong baselines*: CLIP based fine-tuning (Kim et al., 2022) with extra semantic supervision from text guidance. It is worth noting that many fine-tuning works (Kim et al., 2022; Kwon et al., 2023) have been proposed in the context of data editing and image transferring within the same or related domains (e.g, change a smiling face to non-smiling one), which is an easier case with much smaller domain gap compared to this work. For such tuning based methods, it is possible to either directly sample from the tuned models, or adopt a reference image to perform domain transfer. The latter represents a relatively easier case as it bypasses the sampling stage. However, as shown in Sec. 4.2 and Fig. 5 in the appendices, neither works in this proposed few-shot generalization scenario. In addition, we also note that several recent works (Smith et al., 2023) start to deploy the LoRA (Hu et al., 2021a) based tuning for text-to-image diffusion models, however, as suggested by recent analysis (Biderman et al., 2024; Kwon et al., 2024), full fine-tuning generally outperforms LoRA in terms of performance and sampling efficiency both in LLMs and CV.

**Resource Budget and Implementation.** We use $N = 1000$ images for OOD domains, and set the approximate tuning time for 30 minutes to ensure the fair comparison for baseline methods. For deterministic diffusion, we adopt the standard DDIMs skipping step technique to accelerate both processes using 60 steps in total. Each direction takes an average of 3 - 6 seconds. The geometric optimization tolerance $\omega$ is set to be 0.3 for distance for 0.1 for angles, which leads to a rejection rate of approximately 84.44 % based on initial samples. We use 2 RTX 3090 GPUs for all experiments including baselines. As for CLIP based tuning, we adapt the released code from DiffusionCLIP (Kim et al., 2022), lower the ID preservation and L1 loss to be 0, and increase the default tuning rate from 8e-6 to 1e-5 to coordinate larger domain gaps in this work.

### 4.2 RESULTS, EVALUATIONS AND ANALYSIS

**General Quality for Natural Image Synthesis.** As a general quality evaluation, we calculate the FID scores (Heusel et al., 2017) on 5K generated samples for natural images. The FID scores are averaged over four DDPMs pre-trained on different image domains. As shown in Fig. 1 and Tab. 2, *vanilla tuning* with only image supervisions can *hardly* alter the original generation trajectories and synthesize desired images, always synthesizing in-domain images after comparable tuning time with other tuning baselines. As for methods that finetune the model with additional CLIP loss (Radford et al., 2021), such as DiffusionCLIP (Kim et al., 2022) and Asyrp (Kwon et al., 2023), they relatively perform better for domains closer to their trained domains. Our proposed method shows an opposite trend by achieving better performance in data domains with bigger differences, as it is easier to avoid mode interference with larger domain gaps in the latent spaces.

**Data Diversity for Natural Images Synthesis.** Data diversity is another coomon evaluation criteria in addition to the general quality in natural image synthesis. In Fig. 4, we qualitatively show examples of the reference and sampled data after denoising for OOD synthesis, as well as the top 5 nearest raw images from the overall OOD samples. The LPIPS scores (Zhang et al., 2018) between the generated images and their nearest neighbors are $0.49 \pm 0.03$ and $0.47 \pm 0.06$ for sketch and

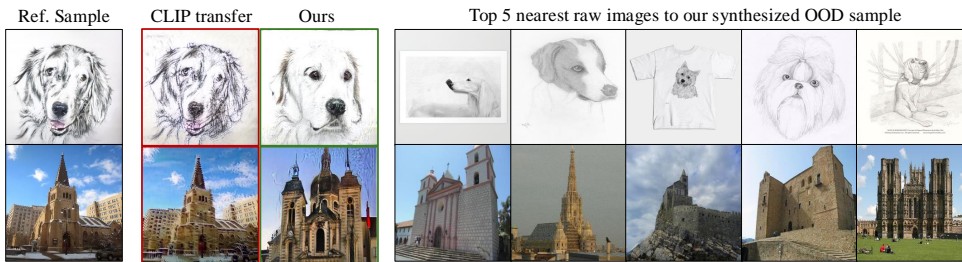

**Ref. Sample** **CLIP transfer** **Ours** Top 5 nearest raw images to our synthesized OOD sample

Figure 4: **Generated OOD images (in green boxes) of our *DiscoveryDiff* are different from the latent reference and the top 5 nearest neighbors raw images.** In contrast, direct transfer from an OOD reference using tuned models fails to synthesize new images (in red boxes).

church OOD domains, respectively. We acknowledge that the diversity of OOD samples is not yet perfect, but they establish sufficient visual differences to be distinguishable from the references.

**Astrophysical Evaluations.** Unlike natural images, evaluations of scientific data usually follow their established evaluation protocols on specific tasks, neither general quality computed on FID nor the data diversity are applicable in this case. Therefore, we perform the subjective evaluation in a *non cherry-picked manner*, and ask astrophysicists to rate the quality using the Mean Opinion Scores (MOS) of a scale between 1-5, with a maximum score of 5 with respect to the ground truth observation and simulation data. In general, the quality of galaxy images is assessed based on whether they contain meaningful morphological information. For radiation data, the evaluation is conducted independently via the comparison with the simulated physical distribution after transferring back to different wavelength domains measures in $\mu$m, ranging from $10^{-1}$ to $10^8$. Our proposed *DiscoveryDiff* achieves better performance in both astrophysical datasets compared to the strong baseline with CLIP tuning and reference image. More details can be found in Appendix E.1.

For other discussions on the impact of N, the latent steps for inversion, and the optimization tolerance, please refer to Appendix E.2.

## 5 RELATED WORKS

This work is closely related to several research fields such as the *generalization ability of generative models* (Wang et al., 2022a; Rombach et al., 2022), the *diffusion models and their deterministic variants* (Sohl-Dickstein et al., 2015; Song & Ermon, 2019; Song et al., 2021; 2023), the study on the *latent dynamics and regimes of deep generative models* (Karras et al., 2017; Abdal et al., 2019; Gal et al., 2022; Zhu et al., 2023a; Nitzan et al., 2023), as well as recent downstream applications that use diffusion models for scientific explorations in a data-rare cases (Xu et al., 2023a;b). Our work adopts a few-shot scenario to study the generalization abilities, uses the deterministic variant as the tool to achieve a bidirectional transition between latent noisy and data spaces, and contributes to a better understanding of those latent spaces. An extended discussion on related work is in Appendix A.

## 6 FURTHER DISCUSSIONS

**Conclusion.** To sum up, we study the domain generalization of DDPMs in the few-shot scenario. From the analytical point of view, we explore the generalization properties of DDPMs on unseen OOD domains. From the methodological perspective, our analytical results allow us to propose a novel paradigm for synthesizing images from new domains without tuning the generative trajectories. We also showcase the superiority of our method in data-sparse cases with large domain gaps.

**Limitations and Broader Impact.** The current limitations and challenge mainly come from the OOD sampling. As previously discussed, the sampling from inverted OOD prior in high dimensionality is a challenging task and open research question, which also directly impact the synthesized image quality. Improved sampling methods are worth investigating as future research directions. This work falls into the category of generative models and their applications, we thus acknowledge that it may pose the same risks of malicious use of synthetic data as other general generative works. However, the primary objective of our work is not performance-driven but to provide a better understanding of the generalization properties of diffusion generative models.

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

We structure the appendices as follows: We provide a detailed discussion on the related work in Appendix A. In Appendix B, we present the background of deterministic diffusion models. In Appendix C, we provide detailed proof for the lemmas used in Sec. 2 of the main paper as part of analytical studies. Meanwhile, Appendix C includes the necessary background on the geometric properties of high-dimensional Gaussians. More latent sampling methods are discussed in Appendix D. Appendix E includes additional details about the generative experiments on unseen OOD domains. We have also included our **core code** as part of the supplementary material.

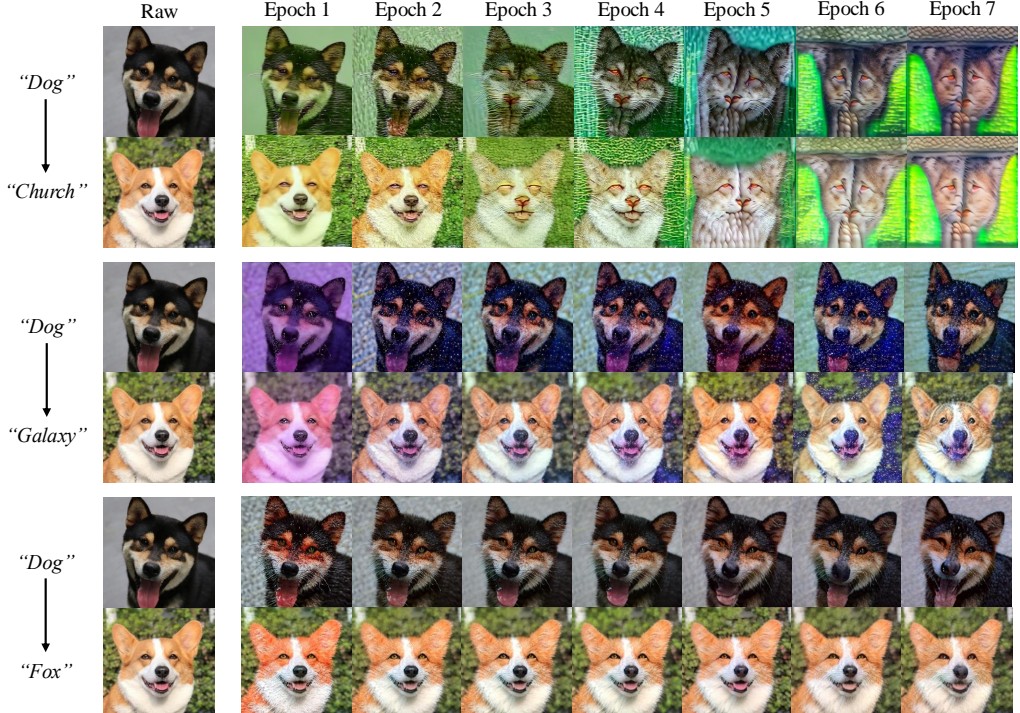

Figure 5: **Fine-tuning methods often fail to transfer the original trained domain to the target OOD domain with large domain gaps.** We qualitatively show how a given ID sample (e.g., a dog RGB image) changes as the tuning epoch increases, using extra CLIP semantic guidance. We note that tuning a pre-trained model to an OOD domain with large gap (e.g., dog-to-church and dog-to-galaxy) usually fails.

## A    RELATED WORK

Due to the space limitations in the main paper, we present a detailed discussion of related work in our appendices.

### A.1    GENERALIZATION IN GENERATIVE MODELS

Domain Generalization (Wang et al., 2022a) that aims to generalize models' ability to extended data distributions has been an important research topic in broad machine learning area (Ganin et al., 2016; Zhao et al., 2020; Zhou et al., 2021; Muandet et al., 2013; Li et al., 2017), with various computer vision applications such as recognition (Peng et al., 2019; Rebuffi et al., 2017), detection (Zhang et al., 2021) and segmentation (Hoffman et al., 2018; Gong et al., 2019). In the vision generative field, it becomes an even more challenging task with the extra demand to sample from the generalized distributions. One popular recent trend in the computer vision community is scaling up the model and dataset sizes as the most intuitive and obvious solutions (Ramesh et al., 2022; Ho et al., 2022a; Rombach et al., 2022). Another scenario to study the domain generalization of generative models is within the few-shot scenario, where we only have a limited amount of data compared to

the training set. In this case, fine-tuning the given model on the limited images Kim et al. (2022) is the most straightforward way to go.

Our work falls into the second category: provided with a pre-trained model and a small set of unseen images different from the model's training domain, we seek to better understand the generalization abilities of DDPMs.

## A.2 DIFFUSION MODELS AND DETERMINISTIC VARIANTS

Diffusion Models (DMs) Sohl-Dickstein et al. (2015); Ho et al. (2020); Song & Ermon (2019) are the state-of-the-art generative models for data synthesis in images (Ramesh et al., 2022; Rombach et al., 2022; Nichol & Dhariwal, 2021; Gu et al., 2022; Dhariwal & Nichol, 2021; Hu et al., 2021b), videos (Ho et al., 2022b), and audio (Kong et al., 2020; Zhu et al., 2023b; Mittal et al., 2021). There are currently two mainstream fundamental formulations of diffusion models, i.e., the denoising diffusion probabilistic models (DDPMs) Ho et al. (2020) and score-based models Song et al. (2020). One common perspective to understand both formulations is to consider the data generation as solving stochastic differential equations (SDEs), which characterize a stochastic process. Based on vanilla models, both branch develops their own deterministic variants, i.e., denoising diffusion implicit models (DDIMs) Song et al. (2021) and consistency models Song et al. (2023), with their core idea to follow the marginal distributions in denoising. Compared to initial DDPMs and Score-based DMs with ancestral sampling, the deterministic variants are solving ODEs instead of SDEs and largely accelerate the generation speed with fewer steps.

We leverage the deterministic variant (DDIMs Song et al. (2021)) as the tool to achieve bidirectional transition between latent noisy space and data space in this work.

## A.3 LATENT SPACE OF DEEP GENERATIVE MODELS

Comprehensive studies of latent space of generative models (Karras et al., 2017; Abdal et al., 2019; Gal et al., 2022) help to better understand the model and also benefit downstream tasks such as data editing and manipulation (Zhu et al., 2016; Shen et al., 2020; Kwon et al., 2023; Zhu et al., 2020). A large portion of work has been exploring this problem within the context of GAN inversion (Xia et al., 2022), where the typical methods can be mainly divided into either learning-based (Zhu et al., 2016; Richardson et al., 2021; Wei et al., 2022; Alaluf et al., 2021) or optimization-based categories (Abdal et al., 2019; 2020; Huh et al., 2020; Creswell & Bharath, 2018). More recently, with the growing popularity of diffusion models, researchers have also focused on the latent space understanding of DMs for better synthesis qualities or semantic control (Rombach et al., 2022; Zhu et al., 2023a; Yang et al., 2024).

Our work also contributes to a better understanding of latent spaces, and aims to introduce a new synthesis paradigm to explore the intrinsic potential of DMs.

## A.4 DIFFUSION MODELS IN SCIENCE

While DMs have been extensively applied in data generation and editing within the multimodal context (Rombach et al., 2022; Ho et al., 2022b; Zhu et al., 2023b; Yang et al., 2024; Zhu et al., 2023a), recent works have extended their application domains to scientific explorations, such as astrophysics (Xu et al., 2023b;a), medical imaging (Kazerouni et al., 2023; Wu et al., 2024a), and biology (Wu et al., 2024b). Compared to conventional computer vision applications, scientific tasks usually exhibit several distinct features. For instance, data acquisition and annotation are generally more expensive due to their scientific nature, resulting in a relatively smaller amount of available data for experiments. Additionally, the evaluation of these works adheres to established conventions within their respective contexts, which are usually different from image synthesis evaluation based on perceptual quality.

Our work also experiments with several astrophysical datasets to showcase the potential of applying our proposed paradigm and method to such specific domains with limited data.

## B    DETERMINISTIC DIFFUSION

Our analytical studies and methodology designs are built upon a specific variant of diffusion formulations, i.e., the deterministic diffusion process. While the original DDPMs involve a stochastic process for data generation via denoising (*i.e.*, the same latent encoding will output different denoised images every time after the same generative chain), there is a variant of diffusion model that allows us to perform the denoising process in a deterministic way, known as the Denoising Diffusion Implicit Models (DDIMs) (Song et al., 2021). DDIMs were initially proposed for the purpose of speeding up the denoising process, however, later research works extend DDIMs from faster sampling application to other usages including the inversion technique to convert a raw image to its arbitrary latent space in a deterministic and tractable way. As briefly stated in our main paper, the core theoretical difference between DDIMs and DDPMs lies within the nature of forward process, which modifies a Markovian process to a non-Markovian one.

The key idea in the context of non-Markovian forward is to consider a family of $\mathcal{Q}$ of inference distributions, indexed by a real vector $\sigma \in \mathbb{R}_{\geq 0}^T$:

$$q_\sigma(\mathbf{x}_{1:T}|\mathbf{x}_0) := q_\sigma(\mathbf{x}_T|\mathbf{x}_0) \prod_{t=2}^{T} q_\sigma(\mathbf{x}_{t-1}|\mathbf{x}_t, \mathbf{x}_0), \tag{5}$$

where $q_\sigma(\mathbf{x}_T|\mathbf{x}_0) = \mathcal{N}(\sqrt{\alpha_T}\mathbf{x}_0, (1-\alpha_T)\mathbf{I})$ and for all $t > 1$,

$$q_\sigma(\mathbf{x}_{t-1}|\mathbf{x}_t, \mathbf{x}_0) = \mathcal{N}(\sqrt{\alpha_{t-1}}\mathbf{x}_0 + \sqrt{1-\alpha_{t-1}-\sigma_t^2} \cdot \frac{\mathbf{x}_t - \sqrt{\alpha_t}\mathbf{x}_0}{\sqrt{1-\alpha_t}}, \sigma_t^2 \mathbf{I}). \tag{6}$$

The choice of mean function from Eqn. 6 ensures that $q_\sigma(\mathbf{x}_t|\mathbf{x}_0) = \mathcal{N}(\sqrt{\alpha_t}\mathbf{x}_0, (1-\alpha_t)\mathbf{I})$ for all $t$, so that it defines a joint inference distribution that matches the "marginals" as desired. The non-Markovian forward process can be derived from Bayes' rule:

$$q_\sigma(\mathbf{x}_t|\mathbf{x}_{t-1}, \mathbf{x}_0) = \frac{q_\sigma(\mathbf{x}_{t-1}|\mathbf{x}_t, \mathbf{x}_0) q_\sigma(\mathbf{x}_t|\mathbf{x}_0)}{q_\sigma(\mathbf{x}_{t-1}|\mathbf{x}_0)}. \tag{7}$$

In fact, in the original paper, the authors also explicitly stated that: " The forward process from Eqn. 7 is also Gaussian (although we do not use this fact for the remainder of this paper)". [5] While this Gaussian property was not emphasized and leveraged in the DDIMs paper, we find it useful in our context to explore the representation and generalization ability of pre-trained DDPMs.

In particular, the hyper-parameters for Gaussian scheduling $\alpha$ and $\beta$ in the context of DDIMs are slightly different from the original formulation in DDPMs (Ho et al., 2020). Denote the original sequences from DDPMs as $\alpha_t'$, then the $\alpha_t$ in this work follows the definition of DDIMs to be $\alpha_t = \prod_{t=1}^{T} \alpha_t'$.

In addition to DDIMs, we note that the score-based formulation has also recently marked a deterministic variant, namely the Consistency Models (Song et al., 2023). The core idea of the consistency model is, to some extent, similar to DDIMs, which allows the vanilla score-based stochastic diffusion models to achieve "one-step" denoising, by following the marginal distributions.

As mentioned in our main paper, the deterministic diffusion is mainly used as a tool in this work for our proposed tuning-free paradigm.

## C    DETAILED PROOFS AND GEOMETRIC PROPERTIES

In this section, we provide detailed proof for the lemmas in Sec. 2. Particularly, Lemma 2.2 is a known property in high-dimensional Gaussian studies.

### C.1    PROOF OF LEMMA 2.1

We restate the lemma below, and provide the detailed proof, which has been introduced in the original DDIM paper (Song et al., 2021).

---

[5]This paper refer to the DDIM paper (Song et al., 2021).

Table 3: **Reconstruction results for arbitrary images via deterministic diffusion.** We use an iDDPM (Nichol & Dhariwal, 2021) trained on AFHQ-Dog and 1K testing OOD images to compute the MAE (mean absolute error) reconstruction metric. Note DDIMs (Song et al., 2021) was initially proposed to accelerate DDPMs sampling, but have not been studied in this OOD reconstruction setting.

| Method | Recons. Domain | MAE ($\downarrow$) |
|---|---|---|
| pSp (Richardson et al., 2021) | CelebA (ID) | 0.079 |
| e4e (Tov et al., 2021) | CelebA (ID) | 0.092 |
| ReStyle (Alaluf et al., 2021) | CelebA (ID) | 0.089 |
| HFGI (Wang et al., 2022b) | CelebA (ID) | 0.062 |
| | Dog (ID) | $0.073 \pm 6e\text{-}4$ |
| | CelebA (OOD) | $0.073 \pm 8e\text{-}4$ |
| DDIMs (Song et al., 2021) | Church (OOD) | $0.074 \pm 8e\text{-}4$ |
| | Bedroom (OOD) | $0.072 \pm 7e\text{-}4$ |
| | Galaxy (OOD) | $0.067 \pm 1e\text{-}3$ |
| | Radiation (OOD) | $0.077 \pm 9e\text{-}4$ |

**Lemma C.1.** *For $q_\sigma(\mathbf{x}_{1:T}|\mathbf{x}_0)$ defined in Eqn. 1 and $q_\sigma(\mathbf{x}_{t-1}|\mathbf{x}_t, \mathbf{x}_0)$ defined in Eqn. 2, we have:*

$$q_\sigma(\mathbf{x}_t|\mathbf{x}_0) = \mathcal{N}(\sqrt{\alpha_t}\mathbf{x}_0, (1 - \alpha_t)\mathbf{I}). \tag{8}$$

*Proof:*
Assume for any $t \le T$, $q_\sigma(\mathbf{x}_t|\mathbf{x}_0) = \mathcal{N}(\sqrt{\alpha_t}\mathbf{x}_0, (1 - \alpha_t)\mathbf{I})$ holds, if:

$$q_\sigma(\mathbf{x}_{t-1}|\mathbf{x}_0) = \mathcal{N}(\sqrt{\alpha_{t-1}}\mathbf{x}_0, (1 - \alpha_{t-1})\mathbf{I}), \tag{9}$$

then we can prove that the statement with an induction argument for $t$ from $T$ to 1, since the base case ($t = T$) already holds.

First, we have that

$$q_\sigma(\mathbf{x}_{t-1}|\mathbf{x}_0) := \int_{\mathbf{x}_t} q_\sigma(\mathbf{x}_t|\mathbf{x}_0)q_\sigma(\mathbf{x}_{t-1}|\mathbf{x}_t, \mathbf{x}_0)d\mathbf{x}_t, \tag{10}$$

$$q_\sigma(\mathbf{x}_t|\mathbf{x}_0) = \mathcal{N}(\sqrt{\alpha_t}\mathbf{x}_0, (1 - \alpha_t)\mathbf{I}), \tag{11}$$

$$q_\sigma(\mathbf{x}_{t-1}|\mathbf{x}_t, \mathbf{x}_0) = \mathcal{N}\left(\sqrt{\alpha_{t-1}}\mathbf{x}_0 + \sqrt{1 - \alpha_{t-1} - \sigma_t^2} \cdot \frac{\mathbf{x}_t - \sqrt{\alpha_t}\mathbf{x}_0}{\sqrt{1 - \alpha_t}}, \sigma_t^2\mathbf{I}\right). \tag{12}$$

According to (Bishop & Nasrabadi, 2006) *2.3.3 Bayes' theorem for Gaussian variables*, we know that $q_\sigma(\mathbf{x}_{t-1}|\mathbf{x}_0)$ is also Gaussian, denoted as $\mathcal{N}(\mu_{t-1}, \Sigma_{t-1})$ where:

$$\mu_{t-1} = \sqrt{\alpha_{t-1}}\mathbf{x}_0 + \sqrt{1 - \alpha_{t-1} - \sigma_t^2} \cdot \frac{\sqrt{\alpha_t}\mathbf{x}_0 - \sqrt{\alpha_t}\mathbf{x}_0}{\sqrt{1 - \alpha_t}} = \sqrt{\alpha_{t-1}}\mathbf{x}_0, \tag{13}$$

$$\Sigma_{t-1} = \sigma_t^2\mathbf{I} + \frac{1 - \alpha_{t-1} - \sigma_t^2}{1 - \alpha_t}(1 - \alpha_t)\mathbf{I} = (1 - \alpha_{t-1})\mathbf{I}. \tag{14}$$

Therefore, $q_\sigma(\mathbf{x}_{t-1}|\mathbf{x}_0) = \mathcal{N}(\sqrt{\alpha_{t-1}}\mathbf{x}_0, (1 - \alpha_{t-1})\mathbf{I})$, which allows to apply the induction argument.

*Q.E.D*

## C.2 PROOF OF LEMMA 4.2

**Lemma C.2.** *Mixtures of spherical Gaussians in $d$ dimensions can be separated provided their centers are separated by more than $d^{\frac{1}{4}}$ distance (i.e, a separation of $\Omega(d^{\frac{1}{4}})$), and even by $\Omega(1)$ separation with more sophisticated algorithms.*

*Proof:*

According to existing established understanding (Lemma 2.8 from Blum et al. (2020)), for a $d$-dimensional spherical Gaussian of variance 1, all but $\frac{4}{c^2}e^{-\frac{c^2}{4}}$ fraction of its mass is within the annulus $\sqrt{d-1} - c \leq r \leq \sqrt{d-1} + c$ for any $c > 0$, as illustrated in Fig. 6.

Given two spherical unit variance Gaussians, we have most of the probability mass of each Gaussian lies on an annulus of width $O(1)$ at radius $\sqrt{d-1}$. Also, $e^{-|x|^2/2}$ factors into $\prod_i e^{-x_i^2/2}$ and almost all of the mass is within the slab $\{\mathbf{x}| - c \leq x_1 \leq c\}$, for $c \in O(1)$.

Now consider picking arbitrary samples and their separability. After picking the first sample $\mathbf{x}$, we can rotate the coordination system to make the first axis point towards $\mathbf{x}$. Next, independently pick a second point $\mathbf{y}$ also from the first Gaussian. The fact that almost all of the mass of the Gaussian is within the slab $\{\mathbf{x}| - c \leq x_1 \leq c, c \in O(1)\}$ at the equator says that $\mathbf{y}$'s component along $\mathbf{x}$'s direction is $O(1)$ with high probability, which indicates $\mathbf{y}$ should be nearly perpendicular to $\mathbf{x}$, and thus we have $|\mathbf{x} - \mathbf{y}| \approx \sqrt{|\mathbf{x}|^2 + |\mathbf{y}|^2}$.

More precisely, we note $\mathbf{x}$ is at the North Pole after the coordination rotation with $\mathbf{x} = (\sqrt{(d)} \pm O(1), 0, ...)$. At the same time, $\mathbf{y}$ is almost on the equator, we can further rotate the coordinate system so that the component of $\mathbf{y}$ that is perpendicular to the axis of the North Pole is in the second coordinate, with $\mathbf{y} = (O(1), \sqrt{(d)} \pm O(1), ...)$. Thus we have:

$$(\mathbf{x} - \mathbf{y})^2 = d \pm O(\sqrt{d}) + d \pm O(\sqrt{d}) = 2d \pm O(\sqrt{d}), \tag{15}$$

and $|\mathbf{x} - \mathbf{y}| = \sqrt{(2d)} \pm O(1)$.

Given two spherical unit variance Gaussians with centers $\mathbf{p}$ and $\mathbf{q}$ separated by a distance $\delta$, the distance between a randomly chosen point $\mathbf{x}$ from the first Gaussian and a randomly chosen point $\mathbf{y}$ from the second is close to $\sqrt{\delta^2 + 2d}$, since $\mathbf{x}-\mathbf{p}$, $\mathbf{p}-\mathbf{q}$, and $\mathbf{q}-\mathbf{y}$ are nearly mutually perpendicular, with:

$$|\mathbf{x} - \mathbf{y}|^2 \approx \delta^2 + |\mathbf{z} - \mathbf{q}|^2 + |\mathbf{q} - \mathbf{y}|^2 = \delta^2 + 2d \pm O(\sqrt{d}). \tag{16}$$

To ensure that the distance between two points picked from the same Gaussian are closer to each other than two points picked from different Gaussians requires that the upper limit of the distance between a pair of points from the same Gaussian is at most the lower limit of distance between points from different Gaussians. This requires the following criterion to be satisfied:

$$\sqrt{2d} + O(1) \leq \sqrt{2d + \delta^2} - O(1), \tag{17}$$

which holds when $\delta \in \Omega(d^{1/4})$.

Thus, mixtures of spherical Gaussians can be separated provided their centers are separated by more than $d^{1/4}$.

*Q.E.D*

## C.3 GEOMETRIC PROPERTIES

We consistently observe three geometric properties for the inverted OOD latent encodings. We provide a more detailed discussion on what each property implies in this sub-section.

Recall the three geometric properties as below:

***Observation 1:*** *For any OOD sample pairs $\mathbf{x}_{inv,i}^{out}$ and $\mathbf{x}_{inv,j}^{out}$ from the sample set, the Euclidean distance between these two points is approximately a constant $d_o$.*

***Observation 2:*** *For any three OOD samples $\mathbf{x}_{inv,i}^{out}$, $\mathbf{x}_{inv,j}^{out}$ and $\mathbf{x}_{inv,k}^{out}$ from the sample set, the angle formed between $\overrightarrow{\mathbf{x}_{inv,k}^{out}\mathbf{x}_{inv,i}^{out}}$ and $\overrightarrow{\mathbf{x}_{inv,k}^{out}\mathbf{x}_{inv,j}^{out}}$ is always around $60°$.*

***Observation 3:*** *For any OOD sample pairs $\mathbf{x}_{inv,i}^{out}$ and $\mathbf{x}_{inv,j}^{out}$ from the sample set, let $O$ denote the origin in the high-dimensional space, the angle formed between $\overrightarrow{O\mathbf{x}_{inv,i}^{out}}$ and $\overrightarrow{O\mathbf{x}_{inv,j}^{out}}$ is always around $90°$.*

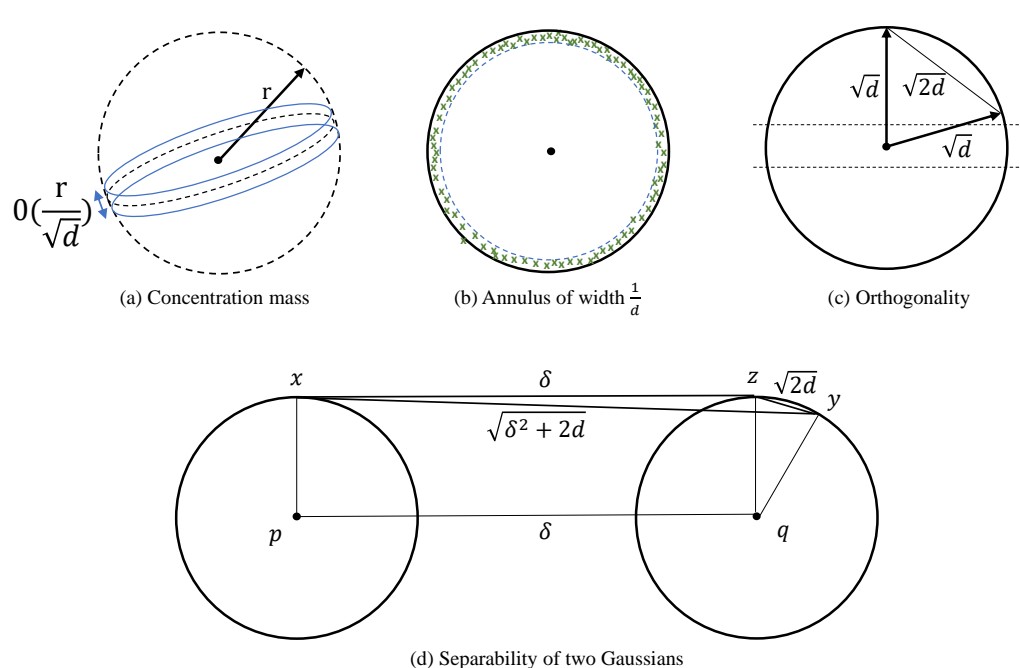

(a) Concentration mass   (b) Annulus of width $\frac{1}{d}$   (c) Orthogonality

(d) Separability of two Gaussians

Figure 6: **Illustration of various geometric properties of high-dimensional Gaussians.** (a) and (b) show the probability concentration mass is mainly centered around a thin annulus around the equator. (c) illustrates the geometric observation on the orthogonality of sample pairs. (d) illustrates the idea of separating two Gaussian distributions in high-dimensional spaces.

For the first observation, when the sample pairs keep approximately the same distance, the direct implication is that those samples are likely to be drawn from some convex region in the high-dimensional space (Wang, 2012). One typical example is the spherical structure, where every data points exhibit an equal distance from the center.

The second geometric property suggests that the unknown samples could lie on a regular lattice near a low-dimensional manifold or sub-manifold, where the local geometry of the manifold is approximately Euclidean. However, a less evident implication is that for samples drawn from a high-dimensional Gaussian, this property also holds, as detailed in the next section C.4, and illustrated in Fig. 6(c).

The third geometry property implies that the sample points might be isotropic in nature, who are rotationally symmetric around any point in the space. Therefore, any two points drawn from the distribution are equally likely to lie along any direction in the space. This property is also observed for a high-dimensional Gaussian (Blum et al., 2020), whose covariance matrix is proportional to the identity matrix.

We acknowledge that to deduce a distribution in high-dimensional space solely based on its geometric properties is very challenging, and there may exist other complex distributions that exhibit similar properties we have observed. However, combined with our theoretical analysis and empirical observations, the OOD Gaussian assumption seems to hold well. Explicitly, we find the above geometric properties do not hold for images $x_0$ from the data space. For instance, the angle of samples to the origin is approximately $75°$ rather than $90°$.

## C.4 HIGH-DIMENSIONAL GAUSSIAN

Gaussian in high-dimensional space establishes various characteristic behaviors that are not obvious and evident in low-dimensionality. A better understanding of those unique geometric and proba-

bilistic behaviors is critical to investigate the latent spaces of DDMs, since all the intermediate latent spaces along the denoising chain are Gaussian as demonstrated and proved in our previous sections.

We present below several properties of high-dimensional Gaussian from (Blum et al., 2020), note those are known and established properties, we therefore omit the detailed proofs in this supplement, and ask readers to refer to the original book if interested.

**Property D.1.** *The volume of a high-dimensional sphere is essentially all contained in a thin slice at the equator and is simultaneously contained in a narrow annulus at the surface, with essentially no interior volume. Similarly, the surface area is essentially all at the equator.*

This property above is illustrated in Fig. 6(a)(b), where the sampled ID encodings are presented in a narrow annulus.

**Lemma D.2.** *For any $c > 0$, the fraction of the volume of the hemisphere above the plane $x_1 = \frac{c}{\sqrt{d-1}}$ is less than $\frac{2}{c} e^{-\frac{c^2}{2}}$.*

**Lemma D.3.** *For a d-dimensional spherical Gaussian of variance 1, all but $\frac{4}{c^2} e^{-c^2/4}$ fraction of its mass is within the annulus $\sqrt{d-1} - c \leq r \leq \sqrt{d-1} + c$ for any $c > 0$.*

The lemmas above imply that the volume range of the concentration mass above the equator is in the order of $O(\frac{r}{\sqrt{d}})$, also within an annulus of constant width and radius $\sqrt{d-1}$. In fact, the probability mass of the Gaussian as a function of $r$ is $g(r) = r^{d-1} e^{-r^2/2}$. Intuitively, this states the fact that the samples from a high-dimensional Gaussian distribution are mainly located within a manifold, which matches our second geometric observation.

**Lemma D.4.** *The maximum likelihood spherical Gaussian for a set of samples is the one over center equal to the sample mean and standard deviation equal to the standard deviation of the sample.*

The above lemma is used as the theoretical justification for the proposed empirical search method in (Zhu et al., 2023a). We also adopt the search method using the Gaussian radius for identifying the operational latent space along the denoising chain to perform the OOD sampling.

**Property D.5.** *Two randomly chosen points in high dimension are almost surely nearly orthogonal.*

The above property corresponds to the *Observation 3*, where two inverted OOD samples consistently form a 90° angle at the origin.

## D    MORE DEATILS ABOUT THE LATENT SAMPLING METHODS

We present here the detailed algorithms for our proposed latent sampling methods, and discuss many other sampling methods that we have tested during experiments.

### D.1    ALGORITHMS

While we have described the procedure of our proposed latent sampling method in Sec. 3, the following Algo. 2 includes details of the geometric optimizations.

### D.2    OTHER SAMPLING METHODS

In addition to the main sampling method introduced in the main paper, we have tested many other sampling methods for mining the qualified OOD latent encodings.

We list those sampling methods below as extra information and provide a brief discussion for each.

**Approach 1: Estimated Gaussian Sampling.** An intuitive way to achieve the OOD sampling based on our analytical understanding from Sec. 2 is to directly fit the latent encodings with a Gaussian distribution and then sample from the estimated Gaussian. However, we note that the high-dimensional Gaussian estimation itself remains as a challenging and complex research topic, especially in a multi-variant case (Zhou et al., 2011; Bai & Shi, 2011). In general, a reliable estimation of means and variances requires data samples at least 10 times the dimensionality, known as the *"rule of ten"* (Johnson et al., 2002), which contradicts our few-shot setup. Empirically, we also observe that

---

**Algorithm 2** Latent Sampling with Geometric Optimizations

---

**Input:** A sampled OOD latent encoding $x'_{od,t}$, $N$ inverted OOD latent $\{\mathbf{x}^1_{od,t}, \mathbf{x}^2_{od,t}, ..., \mathbf{x}^N_{od,t}\}$, distance tolerance $\omega_d$, angle tolerance $\omega_a$, $N_{ref}$ OOD reference samples
**Output:** True or False
*// Step 1: get pair-wise distance from the inverted OOD encodings $\mathbf{x}_{od,t}$*
$d_{od} \leftarrow 0$
**for** $i = 1, 2, ..., n$ **do**
    $(p, q) \leftarrow$ RandomInt$(0, N - 1)$
    $d_{od}$ += $Euclidien\_distance(x^p_{od,t}, x^q_{od,t})$
**end for**
$d_{od} \leftarrow d_{od}/n$
*// Step 2: Geometric optimization based on pre-defined tolerances*
**for** $i = 1, ..., N_{ref}$ **do**
    $d \leftarrow Euclidien\_distance(x^i_{od,t}, x'_{od,t})$
    **if** $d < d_{od} - \omega_d$ or $d > d_{od} + \omega_d$ **then**
        return $False$
    **end if**
**end for**
**for** $i = 1, ..., N_{ref}$ **do**
    $(p, q) \leftarrow$ RandomInt$(0, N_{ref} - 1)$
    $\varphi \leftarrow Angle(\overrightarrow{x'_{od,t}x^p_{od,t}}, \overrightarrow{x'_{od,t}x^q_{od,t}})$
    **if** $\varphi < 60 - \omega_a$ or $\varphi > 60 + \omega_a$ **then**
        return $False$
    **end if**
**end for**
return $True$

---

the synthesis quality with vanilla Gaussian sampling is not very promising. The key reason for this is the gap between the theoretical foundation and practical model training, as also discussed in our main paper.

**Approach 2: MCMC Sampling.** As an improved statistical method over vanilla Gaussian sampling, we also tested the MCMC sampling technique, which should provide a better and more precise estimation of the distribution based on the inverted latent samples. However, one practical challenge we encountered during the experiments is that MCMC sampling takes an extremely long time for high-dimensional data in this context (i.e., several days using Metropolis-Hastings and Gibbs, which even exceeds the time required to tune the entire model). Therefore, we do not recommend or include this method in our main paper.

**Approach 3: Gaussian Sampling w/ Geometric Optimization.** A more practical implementation of the Gaussian sampling is to leverage the geometric properties as the domain-specific criteria to further optimize the quality of latent encodings, just as described in Algo. 2. We note this improves over the vanilla Gaussian sampling, but still qualitatively suffers from the mode interference issue.

# E MORE DETAILS FOR GENERATIVE EXPERIMENTS

## E.1 BACKGROUND AND EVALUATION ABOUT THE ASTROPHYSICAL DATA

**Galaxy Data.** The images from the GalaxyZoo dataset (Willett et al., 2013) are observation data of galaxies that belong to one of six categories - elliptical, clockwise spiral, anticlockwise spiral, edge-on, star/don't know, or merger. The original data format of those galaxy images are also RGB images, thus "somewhat" similar to natural images, but they contain important morphological information to study the galaxies in astronomy.

The evaluation of the synthesized galaxy data is based on the expertise of astrophysicists if they could reliably classify the generated images into one of the known categories.

**Radiation Data.** For the radiation data from (Xu et al., 2023a), the original format is physical quantity instead of RGB images, which correspond to the dust emission.

Dust is a significant component of the interstellar medium in our galaxy, composed of elements such as oxygen, carbon, iron, silicon, and magnesium. Most interstellar dust particles range in size from a few molecules to 0.1 mm (100 m), similar to micrometeoroids. The interaction of dust particles with electromagnetic radiation depends on factors like their cross-section, the wavelength of the radiation, and the nature of the grain, including its refractive index and size. The radiation process for an individual grain is defined by its emissivity, which is influenced by the grain's efficiency factor and includes processes such as extinction, scattering, absorption, and polarization.

In RGB images of dust emission, different colors represent emissions at three wavelengths: blue for 4.5 $\mu$m, green for 24 $\mu$m, and red for 250 $\mu$m. The blue color typically indicates short-wavelength dust emission from point sources, such as young stars or young stellar objects. The green color represents mid-wavelength dust emission from warm and hot dust. The red color signifies long-wavelength dust emission from cold dust.

Warm/hot dust emission (green) is usually found around stars, which appear as blue-colored dots. Since warm dust often mixes with cold dust on the outer edges of bubble structures, the resulting color is often yellowish. Cold dust extends farther from the stars, giving the background or areas outside star clusters a red appearance. In the case of massive star clusters, stellar feedback, such as radiation and stellar winds, can blow away the surrounding gas and dust, creating black or blank areas. Typically, RGB images show more extensive red emission with some orange/yellow emission, displaying filamentary and bubble structures, along with blue and/or white dotted point source emissions.

The above background is considered as part of the underlying evaluation criteria when performing subjective evaluation on the quality of generated radiation data.

### E.2 More Experimental Results

We provide extended discussions in this section for the readers who are interested in more subtitle experimental details.

#### E.2.1 Discussion on the Latent Step $t$, Stochasticity and Mode Interference

In our main paper, we briefly discuss the impact of the latent diffusion step $t$ where we perform the inversion and OOD latent sampling. While we empirically find that $t \approx 800$ is a reasonable range for the choice of $t$, we note there exists an entangled mechanism for the trade-off between the sampling difficulty and the mode interference issue.

For the diffusion step $t$, recent studies (Zhu et al., 2023a; Yang et al., 2024) suggest that $t$ characterizes the formation of image information at different stages of the denoising process. Intuitively, the early stage of the denoising process (e.g., $t > 800$) represents a rather chaotic process, the mixing step $t_m$ (Zhu et al., 2023a) signifies a critical stage where the image semantic information starts to form, and the later stage where $t$ is close to 0 demonstrates a stage during which more fine-grained pixel-level information are introduced to the final generated data. From the distribution point of view, the influence of $t$ can be interpreted as the convergence of distributions, where $t = T$ is a standard Gaussian by definition, thus the ID and OOD modes are more difficult to separate. However, as the denoising process gets closer to the real image space at $t = 0$, the sampling difficulty increases as the implicit distribution moves away from the standard Gaussian.

Meanwhile, the diffusion step $t$ is not the only factor that impacts the trade-off between sampling difficulty and mode interference. While scarcely discussed in the main paper, we note the stochasticity of the denoising trajectory also plays a similar role as the diffusion step in this work. The stochasticity of the denoising trajectory in DMs has been proven to be generally beneficial in improving the synthesis quality (Karras et al., 2022; Kim et al., 2022; Kwon et al., 2023; Zhu et al., 2023a). In this work, while we choose the $\eta = 0$ for the main paper, a tolerance for a certain range of stochasticity allows us to follow a "relatively deterministic" denoising process $p_{\eta=k}$, with $k \neq 0$, instead of the completely deterministic $p_i$. We hereby refer to it as "bandwidth of the unseen trajectories," denoted as $\mathcal{B}_{\eta,t}$, which can be used to quantify the "mode interference". Another interpretation is to analog the trajectory bandwidth $\mathcal{B}_{\eta,t}$ to the actual subspace volume occupied by the OOD latent samples.

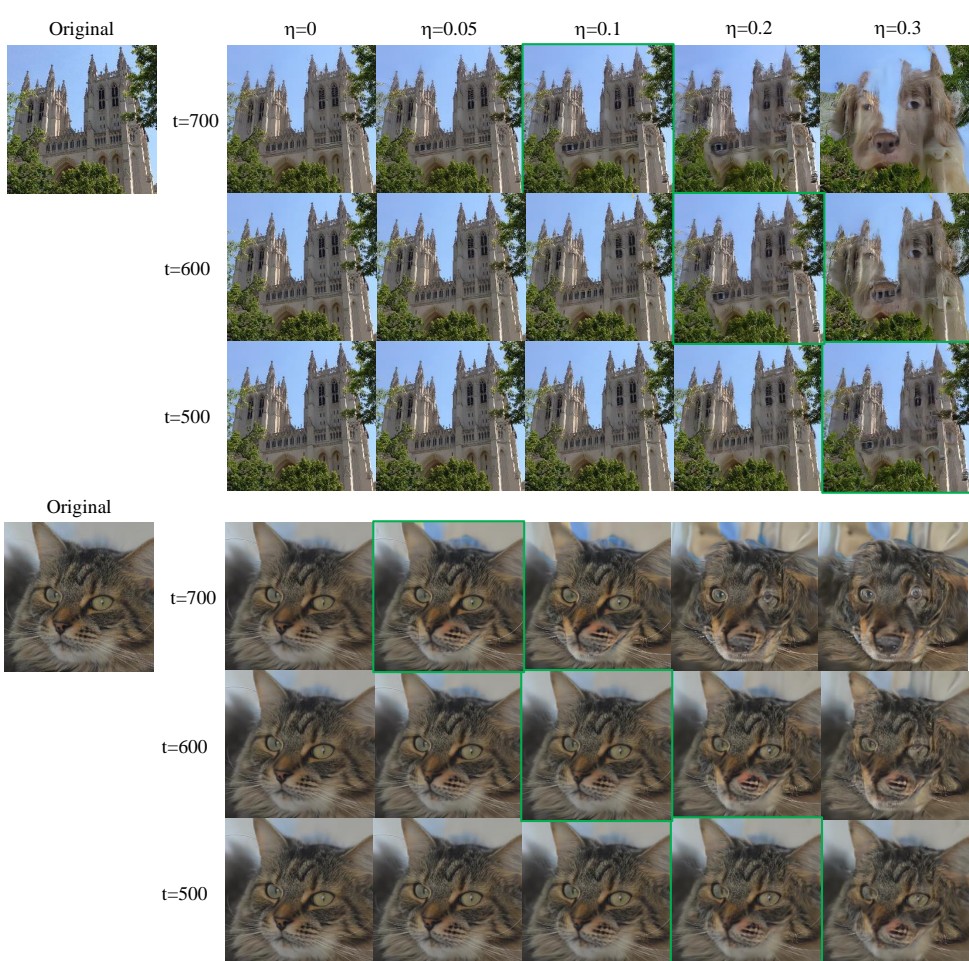

Figure 7: **Illustration of unseen trajectory bandwidth at different diffusion steps.** We show qualitative examples using the iDDPM (Nichol & Dhariwal, 2021) trained on AFHQ-Dog-256 as the base model, the examples of church and cat are both unseen domain images. The image in green boxes indicates the bandwidth we have empirically selected to preserve the reconstruction quality. Compared to the trained image domain (*i.e.*, *dogs*), *cats* have a smaller domain gap than *churches*. Different from the conventional understanding that a smaller domain gap is beneficial for better and easier generalization from a trained model, we observe a larger domain gap signifies a larger bandwidth, making it easier to perform the OOD sampling and synthesis.

Fig. 7 shows more qualitative results for the bandwidth search in the reconstruction task and reveals its connection to the diffusion step $t$. Overall, the bandwidth is a hyper-parameter that relates to the base model and the unseen domains, and the diffusion step $t$, while the bandwidth gets larger at the latent spaces closer to the raw image domains, sampling from OOD unseen distributions also gets more difficult.

### E.2.2 DISCUSSION ON MODEL DESIGNS

Among four base DDPMs we have tested, there are two architecture variants namely the improved DDPM (Nichol & Dhariwal, 2021) and vanilla DDPM (Ho et al., 2020). The difference between the two variants lies within the scheduler design for the Gaussian perturbation kernels: improved DDPM uses a cosine scheduler while vanilla DDPM adopts a linear one. Our experiments suggest that iDDPM in general synthesizes images with better quality in terms of FID scores, which aligns

with previous studies (Nichol & Dhariwal, 2021; Zhu et al., 2023a). One implication from the above observation is that the domain generalization abilities studied in this context is inherited from the performance of model's original performance.

### E.2.3 DISCUSSION ON THE NUMBER OF OOD IMAGES AND REJECTION CRITERIA

While increasing the number of OOD raw images is generally beneficial, there is always a trade-off between resource requirements and performance. Given the constraints of our experimental setup, we selected $N = 1000$ for our experiments. In practice, we find that the number of OOD samples, $N$, is quite robust for computing geometric properties across different base diffusion models, with values ranging from 800 to 1200.

Regarding the rejection criteria, there is a trade-off between performance and the rejection rate, which depends on how different OOD domains behave in inverted latent spaces. While stricter criteria result in a higher rejection rate, we find that a distance tolerance between 0.2 and 0.3, along with an angle tolerance around 0.1, are reasonable empirical choices.

### E.2.4 MORE QUALITATIVE RESULTS

In particular, we present qualitative examples from tuning-based methods in Fig. 5 and observe that these methods often fail when there is a relatively large gap between the target OOD domain and the original trained domain.

More synthesized examples of our proposed method are included in Fig. 8. We also show part of the raw natural image samples used in our work in Fig. 9, Fig. 10, and Fig. 11, which helps to evaluate the diversity of the generated data.

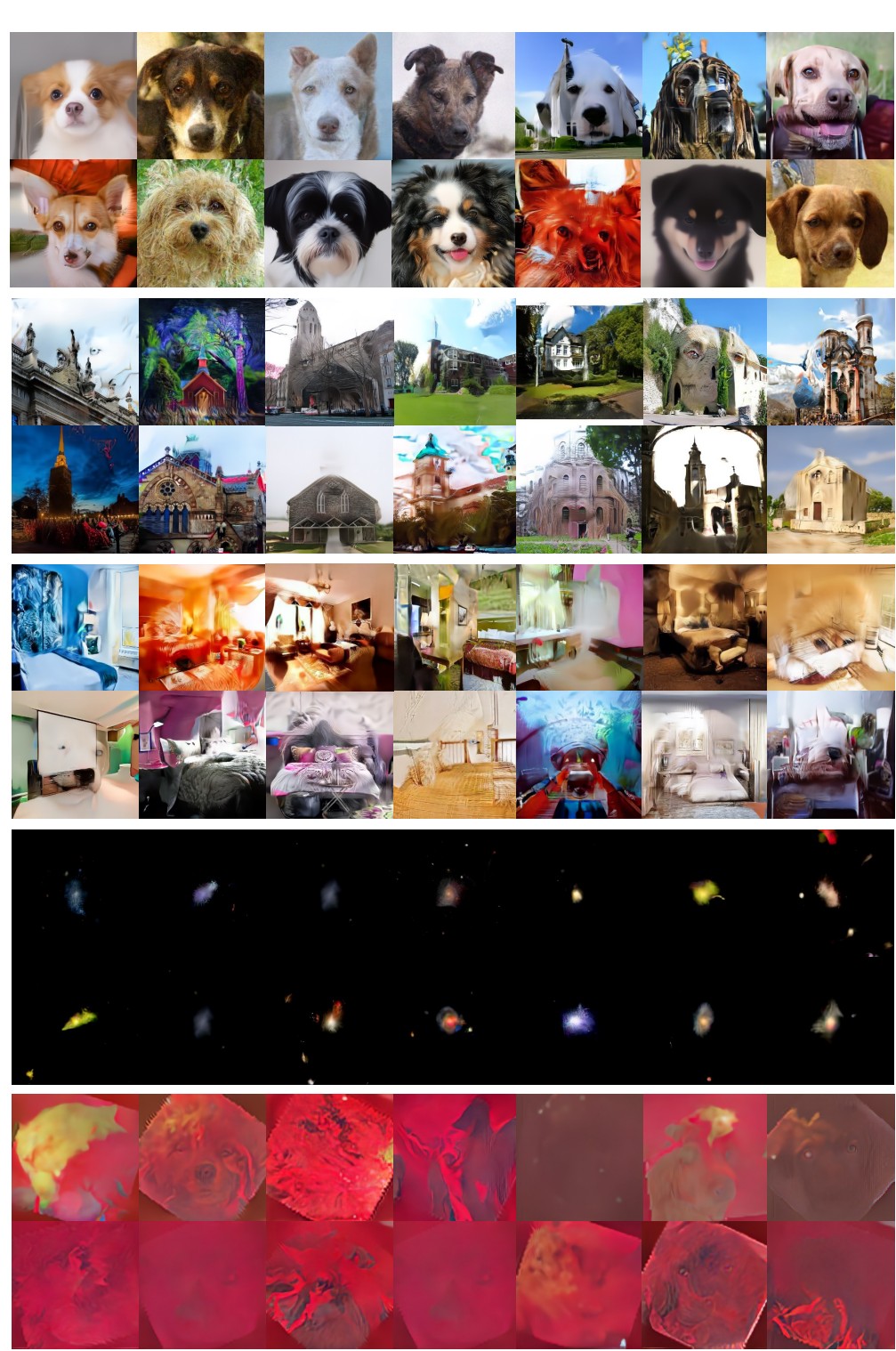

Figure 8: **Additional qualitative results from our proposed method for synthesizing OOD data without tuning the model parameters.**

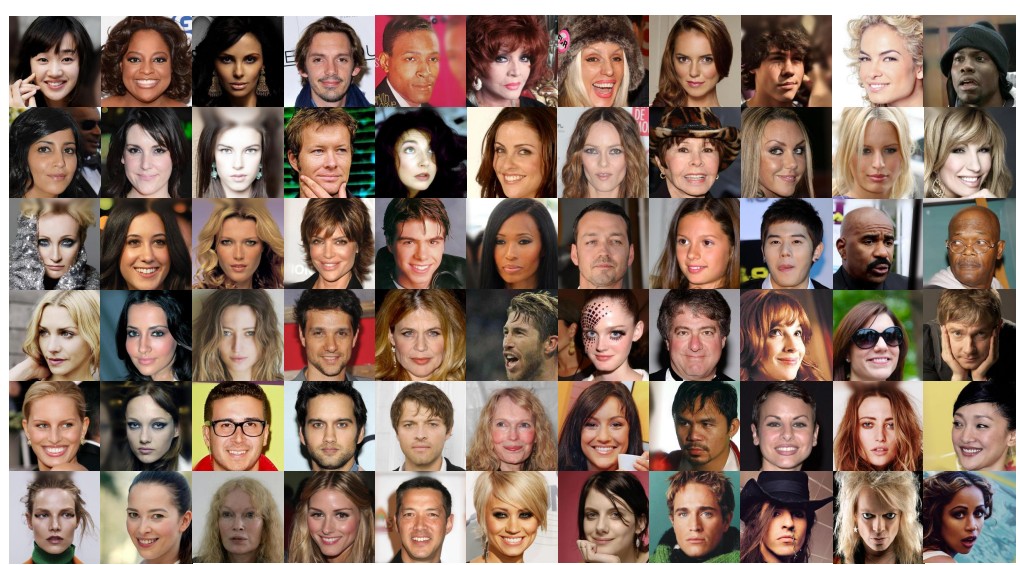

Figure 9: Examples of raw human face images used as OOD samples.

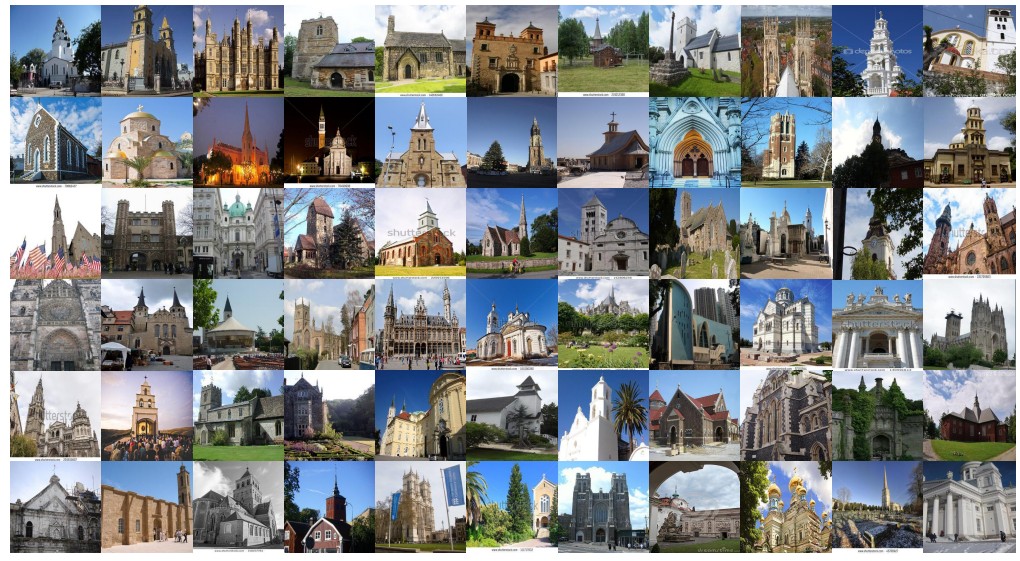

Figure 10: Examples of raw church images used as OOD samples.

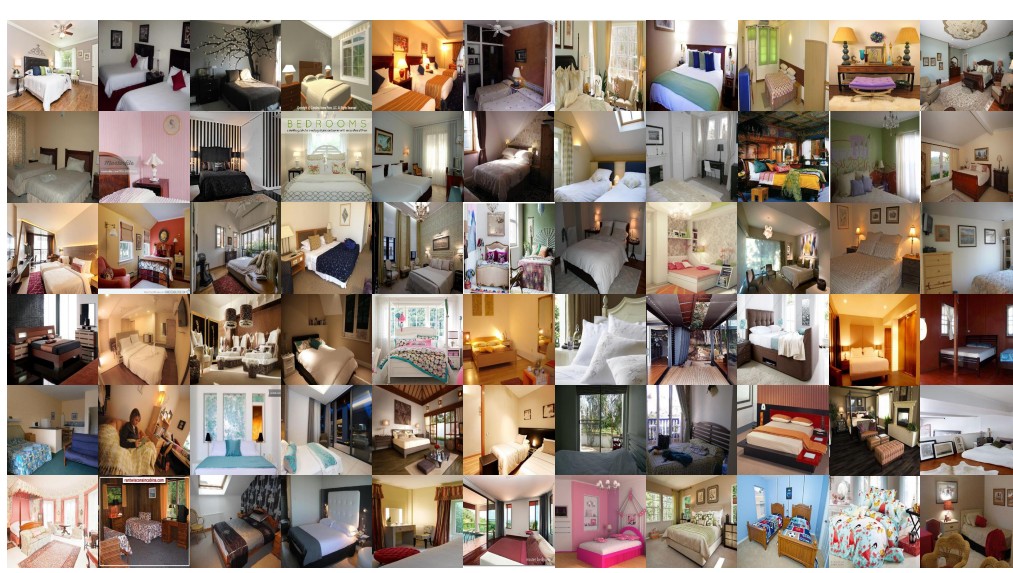

Figure 11: Examples of raw bedroom images used as OOD samples.

