# OpenReview forum: "Discovery and Expansion of New Domains within Diffusion Models"
_ICLR.cc/2025/Conference — ICLR 2025 Conference Withdrawn Submission_

### Official Review · Reviewer_sJLR · 2024-10-31

**Soundness:** 2
**Presentation:** 3
**Contribution:** 2
**Rating:** 5
**Confidence:** 4

**Summary:**

This paper explores a tuning-free approach to generalizing pre-trained Diffusion Probabilistic Models (DDPMs) for out-of-domain (OOD) data synthesis in few-shot scenarios. By leveraging high-dimensional Gaussian priors, spherical interpolation, and geometric optimizations, the method avoids "mode interference" and expands DDPM applications to domains like scientific imagery and astrophysical data in few-shot settings. Extensive experiments demonstrate that this approach effectively synthesizes diverse, high-quality images, offering a new pathway for OOD generation in resource-limited and interdisciplinary settings (based-on domain-specific pre-trained models).

**Strengths:**

1. The paper introduces a paradigm for generating out-of-domain (OOD) data without modifying model parameters, allowing effective cross-domain generalization. This reduces computational costs, making it feasible for resource-limited scenarios.

2. Through geometric optimization and spherical interpolation, the method effectively mitigates "mode interference," ensuring that generated OOD samples remain distinct from the original training domain and improving the quality of target domain synthesis.

3. The approach is adaptable to various DDPM variants and applicable to diverse target domains (e.g., scientific imagery, astrophysical data), showing potential for cross-domain and interdisciplinary uses.

**Weaknesses:**

1. The paper also acknowledges that estimating the corresponding OOD prior from few-shot data is extremely challenging. Although certain conclusions from high-dimensional space offer insights for designing the algorithm, the current approach remains complex, combinatorial, and heuristic, and it is difficult to inspire further advancements in new work.

2. If my understanding is correct, the article does not compare an important category of approaches, namely, lightweight fine-tuning of large-scale diffusion models, such as Dreambooth+SD or LoRA+SD. I believe current lightweight fine-tuning methods on SD can now be completed quickly, even on a single GPU within tens of minutes, and typically require only around 10-20 images. Thus, these methods should serve as an important baseline for this article (and any future improvements).

3. I am also curious about the performance of such tuning-free methods on various large-scale diffusion models, such as the SD series, and their sensitivity to different network architectures (e.g., U-Net, DiT, U-ViT, etc.).

**Questions:**

1. Can the complexity, combinatorial nature, and heuristic aspects of the tuning-free approach be simplified or optimized to inspire further improvements in future work?

2. How effective is the tuning-free method on large-scale diffusion models like the SD series?

3. How does the tuning-free approach compare to lightweight fine-tuning methods (e.g., Dreambooth+SD or LoRA+SD), especially considering their efficiency in requiring minimal data and computational resources?

4. How sensitive is the tuning-free method to different network architectures, such as U-Net, DiT, and U-ViT?

5. How sensitive is the tuning-free method to different ode/sde samplers?

---

> ### Comment · Reviewer_sJLR · 2024-11-26
>
> I have read the other reviews and the author's general response. I keep my score and hope the authors to further improve the paper.

---

### Official Review · Reviewer_nazq · 2024-11-01

**Soundness:** 2
**Presentation:** 1
**Contribution:** 2
**Rating:** 3
**Confidence:** 4

**Summary:**

This work introduces a new method for domain adaptation of diffusion models. The main idea is to use DDIM inversion to find latent representations of OOD data. Given such representations, a heuristic method is used to select new latent codes that can result in viable new generations from the OOD domain.

**Strengths:**

- This work tackles an interesting and important topic: the adaptation of diffusion models to other domains, including scientific ones usually omitted by mainstream studies.
- The analysis of the distinction between latent representations of ID and OOD data, given their latent representations provided by reverse DDIM procedure, is insightful.

**Weaknesses:**

- There is a strong statement in line 095: "Very few existing works have explicitly investigated this task," - where the task is diffusion model fine-tuning to different domains. Numerous works are investigating this area - under the name of introducing new concepts, see [1] (with prior-preserving loss that tackles exactly the generalization problem), [2], [3], and many others. See, for example, the GitHub overview: https://github.com/wangkai930418/awesome-diffusion-categorized?tab=readme-ov-file#new-concept-learning. I am missing the clear distinction between few-shot learning of a new concept or styles presented in the plethora of works and "new domain learning" introduced in this submission.
- The presentation of this work needs to be improved. The (almost three pages long) introduction needs to be more accurate, as it highlights several entirely different problems that are not always related to the later results. The overview of related works omits several important positions.
- It is known that the latent representations calculated with the reverse DDIM procedure are not identical to the starting Gaussian noise. This is, by definition, given that the baseline DDIM approach is based on the assumption that the prediction of the noise removed from the image in the t-th backward diffusion step closely approximates the noise of the (t − 1)-th step. This is why several works try to mitigate this issue (see [4,5,6]). Is it possible that the distinction between ID and OOD data comes from the bigger error in the DDIM inversion?
- The presented method requires additional prior knowledge about the target domain (used by Geometrical Optimization) that might be costly to calculate
- The experimental evaluation misses important baselines defined as new concepts learning, as discussed in one of the previous points.
- The generated examples presented in Figure 8 in the appendix seem to be of poor quality. For the astrophysical data, we can see a collapse to a situation where the object of interest is located in the center of an image.

[1] Ruiz, Nataniel, et al. "Dreambooth: Fine tuning text-to-image diffusion models for subject-driven generation." Proceedings of the IEEE/CVF conference on computer vision and pattern recognition. 2023.

[2] Gal, Rinon, et al. "An image is worth one word: Personalizing text-to-image generation using textual inversion." ICLR2023

[3] Qiu, Zeju, et al. "Controlling text-to-image diffusion by orthogonal finetuning." Advances in Neural Information Processing Systems 36 (2023): 79320-79362.

[4] Daniel Garibi, Or Patashnik, Andrey Voynov, Hadar Averbuch-Elor, and Daniel Cohen-Or. Renoise: Real image inversion through iterative noising. arXiv preprint arXiv: 2403.14602, 2024. URL https://arxiv.org/abs/2403.14602v1.

[5] Gaurav Parmar, Krishna Kumar Singh, Richard Zhang, Yijun Li, Jingwan Lu, and Jun-Yan Zhu. Zero-shot image-to-image translation. In ACM SIGGRAPH 2023 Conference Proceedings, pp. 1–11, 2023.

[6] Seongmin Hong, Kyeonghyun Lee, Suh Yoon Jeon, Hyewon Bae, and Se Young Chun. On exact inversion of dpm-solvers. In Proceedings of the IEEE/CVF Conference on Computer Vision and Pattern Recognition, pp. 7069–7078, 2024.

**Questions:**

I am not able to find the discussion about the adaptation to LDM models mentioned in footnote 4 in the appendices. I find it interesting to know if the distinction between OOD and ID data still holds, given that in LDMs "images (latent codes in the LDM autoencoder)" are normalized to follow Normal distribution, so they, and their representations calculated with inverse DDIM are more aligned with initial random gaussian distribution by definition.

---

### Official Review · Reviewer_DvTk · 2024-11-02

**Soundness:** 4
**Presentation:** 2
**Contribution:** 4
**Rating:** 6
**Confidence:** 4

**Summary:**

This paper presents an interesting finding: the latent space created by applying the DDIM Inverse to a pre-trained diffusion model shows unique priors, even for out-of-domain (OOD) images. Building on this observation, the paper introduces DiscoveryDiff, a tuning-free method for generating OOD samples from a few example images. This method is shown to generalize well across OOD distributions and outperforms other few-shot, tuning-based approaches. However, I feel the analysis section is unclear, and I have some questions about certain parts. With a clearer analysis, I believe this paper would be a strong submission.

**Strengths:**

1. This paper presents an interesting insight. In pixel space, different distributions are clustered in complex ways that are difficult to analyze or sample from directly. However, the paper shows that in the latent space created using the DDIM Inverse on a pre-trained diffusion model, these distributions are organized more clearly. This structure allows for effective OOD sampling using only a few example images.
2. The potential applications are valuable. In fields like physics and medical imaging, where generating data is costly, achieving few-shot OOD generation can be particularly useful.
3. The experiments in this paper are strong. I especially appreciate Figure 4, which demonstrates the diversity of generated samples compared to the few-shot examples. This also supports the existence of a "latent Gaussian prior."

**Weaknesses:**

I feel that the analysis section in this paper could be better. Here’s what I understand from it: first, the paper aims to prove the existence of a Gaussian prior (though I don’t fully understand Lemma 2.1; see my question on this). Next, it introduces a “mode interference” phenomenon and suggests that to avoid this, the Gaussian prior needs to be separable. Then, it examines the Gaussian properties and separability of the OOD distribution in latent space. Let me know if I’ve misunderstood anything.

I’m not convinced that the OOD distribution in latent space really approximates a Gaussian, as the proposed sampling method in line 394 also relies on Slerp interpolation. However, I do think there are likely some unique structures for OOD distribution in the latent space. A stronger analysis section could address questions like how these special structures for OOD distribution arise and what might provide a better approximation for this structure than the Gaussian model.

**Questions:**

1. In line 107, the author states that "DDIM exhibits latent Gaussian priors independent from the parameters of trained DDPMs". The author leverages this property for few-shot OOD generation. From the experiments in this paper, I believe these latent Gaussian priors exist, but are dependent on the parameters of trained DDPMs. Since using an initialized, not trained DDPM with DDIM could not achieve few-shot OOD generation. What's the author's opinion towards this independence?

2. Based on my previous hypothesis the latent Gaussian priors depend on the trained DDPM, could the author qualitatively compare the OOD generation using a different pre-trained diffusion model? Will the pretrained dataset affect the latent Gaussian priors? In other words, will the OOD generation have some dramatic change?

3. I do not understand Lemma 2.1. For the diffusion model forward process, we have $p(x_t|x_0) = \mathcal{N}(\sqrt{\alpha_t}x_0, (1 - \alpha_t)I)$. So what is $q_{\sigma}(x_t|x_0)$? What it means when $q_{\sigma}(x_t|x_0) = p(x_t|x_0) = \mathcal{N}(\sqrt{\alpha_t}x_0, (1 - \alpha_t)I)$.

4. For your experiments, within the 5K generation in Table 2, are they all in the target domain? What's the proportion?

---

### Official Review · Reviewer_xee5 · 2024-11-02

**Soundness:** 3
**Presentation:** 3
**Contribution:** 4
**Rating:** 5
**Confidence:** 4

**Summary:**

The paper presents a method for sampling out of distribution (OOD) using diffusion models. The method is based on determining the latent distribution of the OOD under the pushforward of DDIM. Once this distribution has been estimated it is possible to sample from it and run the backwards DDIM to generate new OOD samples. The method has impressive results and provides insights into how to leverage pretrained diffusion models when the data is scarce which would make training a model unfeasible.

**Strengths:**

- The paper presents a novel method for sampling out of distribution
- The method doesn't require extra training and seems to generalize well as shown by the experiments

**Weaknesses:**

- Section 2.3 seems to indicate that the OOD priors will be Gaussian, however the argument is very misleading. The argument relies on looking at the forward process of DDPM, since this one converges to Gaussian they claim that similar things should happen for DDIM. DDIM shares the same law as DDPM if started under the same initial condition, since we are looking at an OOD this is no longer the case and we should not expect to have latent priors
- Some other approaches to sample from the latent OOD are suggested although no explanation or comparison is given. I believe that it would be very fruitful to explain how these methods would be implemented and what results they gave, otherwise it is not worth including as no insights are gained
- The geometric optimization isn't explored in depth, there are no details of the robustness of the method under different hyperpameters or datasets

**Questions:**

See weaknesses, could you please provide more details on robustness of the hyperparameters. As well as explanations of the alternative methods for sampling from the OOD latents.

---

### Author Response · Authors · 2024-11-24
**Overall Responses and Clarifications**

We would like to take this opportunity to thank all the reviewers for their time and effort in reviewing our paper. We appreciate the constructive feedback and suggestions, such as exploring in-depth analysis of other sampling methods and geometric properties, further support for latent OOD priors, and potential simplifications of existing heuristics for future studies. While we have decided to submit the paper elsewhere, we would like to clarify some key points raised in the reviews, as they may be helpful for interested reviewers and future readers.

---
- **Regarding the latent Gaussian prior, its theoretical justification, and the authors’ thoughts on its dependence on model parameters:**

   Recent theoretical works have demonstrated that *hyperparameters and stepwise errors govern the asymptotic behavior of diffusion models and remain bounded over the number of diffusion steps*. Notably, these works establish the capability of pre-trained DDPMs to approximate the MSE-optimal conditional mean estimator (CME) **without requiring convergence to the prior (i.e., the training distribution).** This suggests that the approximation to a Gaussian is embedded in the hyperparameters, and optimality holds even without convergence to the training data, given mild assumptions such as a sufficient number of diffusion steps.

   In our opinion, the independence of the model parameters **primarily impacts empirical outcomes**, such as how modes interfere with one another, without influencing the theoretical asymptotic behavior. This aligns with an intuitive understanding: as Gaussian noise is repeatedly added to a clean data distribution, the distribution will eventually approximate a Gaussian over time, regardless of specific model parameterizations.

---
- **Regarding the difference from “concept learning” in personalized vision tasks:**

   “Concept learning,” as introduced in recent literature following the popularity of large T2I models, focuses on controlling T2I models for fine-grained, personalized purposes. Theoretically, it skews pre-learned large data distributions towards a few provided test-time samples. However, this approach provides no guarantee that the target concept has not already been seen during the training phase of the pre-trained T2I models.

   In contrast, our work specifically aims to study “new distributions” that have never been encountered during training. This is strictly ensured in our experiments through the use of unconditional DMs, guaranteeing that our target distributions are **entirely novel relative to the training data**. Additionally, since many T2I models are not well-suited for scientific use, *we do not consider adding further experimental comparisons with these large T2I models and their subsequent personalized fine-tuning methods.*

---

### Note · Authors · 2024-12-06

I have read and agree with the venue's withdrawal policy on behalf of myself and my co-authors.